# LEARNING TO ENGINEER PROTEIN FLEXIBILITY

**Petr Kouba**[1,2]  **Joan Planas-Iglesias**[2,3]  **Jiri Damborsky**[2,3]
**Jiri Sedlar**[1]  **Stanislav Mazurenko**[2,3]  **Josef Sivic**[1]

[1]Czech Institute of Informatics, Robotics and Cybernetics, Czech Technical University
[2]Loschmidt Laboratories, Department of Experimental Biology and RECETOX, Masaryk University
[3]International Clinical Research Centre, St. Anne's University Hospital Brno

## ABSTRACT

Generative machine learning models are increasingly being used to design novel proteins for therapeutic and biotechnological applications. However, the current methods mostly focus on the design of proteins with a fixed backbone structure, which leads to their limited ability to account for protein flexibility, one of the crucial properties for protein function. Learning to engineer protein flexibility is problematic because the available data are scarce, heterogeneous, and costly to obtain using computational as well as experimental methods. Our contributions to address this problem are three-fold. First, we comprehensively compare methods for quantifying protein flexibility and identify data relevant to learning. Second, we design and train flexibility predictors utilizing sequential or both sequential and structural information on the input. We overcome the data scarcity issue by leveraging a pre-trained protein language model. Third, we introduce a method for fine-tuning a protein inverse folding model to steer it toward desired flexibility in specified regions. We demonstrate that our method Flexpert-Design enables guidance of inverse folding models toward increased flexibility. This opens up new possibilities for protein flexibility engineering and the development of proteins with enhanced biological activities.

## 1 INTRODUCTION

The goal of this study is to develop a tool that integrates protein flexibility into computational protein design. Computational protein engineering and design are crucial technologies that have shown immense potential in creating next-generation enzymes for a wide range of applications, from the production of fine chemicals, pharmaceuticals, and food ingredients to sustainable, environmentally friendly solutions in green energy and biodegradation (Planas-Iglesias et al., 2021). These approaches aim to propose new amino acid sequences, either from scratch (*de novo*) or by modifying sequences of existing natural proteins to alter their properties in a desired way. Despite the long history of these efforts, protein engineering and design remain labor-intensive and time-consuming, with uncertain outcomes in each case (Listov et al., 2024). Recent advances in generative machine learning have led to several powerful tools for designing proteins with specific topologies (the inverse folding problem), offering the potential for universal protein design platforms (Notin et al., 2024; Kortemme, 2024). However, the number of successful case studies using these tools remains relatively low. One of the major obstacles hindering their broader application is their inability to account for protein flexibility (Chu et al., 2024; Kouba et al., 2023).

Proteins are highly dynamic biomolecules, and the presence of flexible regions is critical for their biological function (Corbella et al., 2023; Lemay-St-Denis et al., 2022). In particular, fine-tuning conformational dynamics of loops close to the active site has been shown to be an important method for modulating substrate specificities (Romero-Rivera et al., 2022), turnover rates (Crean et al., 2021), and pH dependency (Shen et al., 2021) of enzymes. In addition, the biological function of many proteins often requires that a small molecule is transported through their structures, for example via tunnels leading to the active site, whose dynamical properties are thus critical for protein function (Jurcik et al., 2018). Modulating the flexibility of protein parts even far away from the active sites has been shown to be an efficient strategy for improving proteins (Karamitros et al., 2022). However, quantification of protein flexibility remains challenging despite recent progress in both experimental and computational approaches. Experimental methods, such as X-ray crystallography, nuclear magnetic resonance, and hydrogen-deuterium exchange coupled to mass spectroscopy,

require expensive equipment, specialized biophysical expertise, and time-consuming sample preparation and analysis (Peacock & Komives, 2021). Consequently, they lack the high throughput needed for systematic protein flexibility evaluation. Computational methods provide a broad spectrum of approaches for assessing protein flexibility, from rapid coarse-grained modeling to more accurate but highly computationally intensive molecular dynamics (MD) simulations (Kmiecik et al., 2016). However, the available datasets are limited, and systematic comparisons of different methods across large protein datasets are thus lacking. More importantly, it remains unclear how to effectively integrate these computational methods into the state-of-the-art generative tools, increasingly used in protein engineering and design (Kortemme, 2024).

In this work, we aim to fill these gaps by proposing a new method for designing protein flexibility. First, we perform a comprehensive comparison of methods for quantification of protein flexibility and identify relevant data for learning (Section 3). Second, we address the data scarcity issue by proposing new flexibility predictors utilizing just sequential (Flexpert-Seq) or both sequential and structural (Flexpert-3D) information on the input (Section 4.1). Importantly, our goal is to create a fast predictor, which is crucial for its integration into state-of-the-art ML-guided protein engineering pipelines. Third, we introduce a new method (Flexpert-Design) for fine-tuning a protein inverse folding model to make it steerable toward desired flexibility in specified positions (Section 4.2). Our results show that it is possible to predict flexibility from sequence, and that adding structural information further improves the prediction. Furthermore, using our protein flexibility predictor, we demonstrate that inverse folding models can be steered toward generating protein sequences with increased flexibility, suggesting a potential strategy for engineering of protein flexibility.

## 2 RELATED WORK

**Experimental methods for measuring protein flexibility.** Common experimental methods for determining protein flexibility include X-ray crystallography, Nuclear Magnetic Resonance (NMR), and Hydrogen-Deuterium Exchange coupled to Mass Spectroscopy (HDX-MS) (Peacock & Komives, 2021). X-ray crystallography provides flexibility estimates based on the consistency of protein conformations in a crystalline state. The regularity of atomic positions across crystal lattice cells determines the precision of atom location, quantified by the B-factor (or temperature factor), which indicates mobility (Wlodawer et al., 2008). Details on NMR and HDX-MS are given in Appendix A.

**Computational methods for measuring protein flexibility.** Protein flexibility can be predicted using faster, more economical *in silico* methods (Rueda et al., 2007). A common approach involves deriving flexibility from Molecular Dynamics (MD) simulations, which apply Newton's laws to atoms to compute their motion over time. From these simulations, Root Mean Square Fluctuations (RMSF) can be computed per residue to quantify their flexibility as observed in the simulations. While effective, MD is time-consuming, as it must explore a wide range of protein conformations. Elastic Network Models (ENMs) offer a faster alternative by modeling proteins as a system of beads and springs, using Normal Mode Analysis to predict motions (López-Blanco & Chacón, 2016; Tirion, 1996). ENMs were refined by Bahar et al. to include Gaussian and anisotropic movements (Bahar et al., 1997; Atilgan et al., 2001) and are implemented in tools like ProDy (Bakan et al., 2011; Zhang et al., 2021). More details on ENMs are given in Appendix C. Recent machine learning (ML)-based methods, such as AlphaFold (Jumper et al., 2021) and ESMFold (Lin et al., 2022), have been adapted for flexibility prediction. Combining these models with flow matching techniques in AlphaFlow has enabled sampling of broader conformational spaces at about 10 times the speed of MD, though with reduced structural precision (Jing et al., 2024).

**Protein flexibility engineering.** Engineering flexibility in proteins remains a significant challenge. Traditional approaches often involve quantum mechanics (QM), QM/MM, molecular dynamics (MD), and Monte-Carlo simulations, along with extensive experimental testing (Planas-Iglesias et al., 2021). For example, Yu & Dalby (2018) employed MD simulations to analyze correlated motions in enzyme dynamics. Similarly, Osuna's group developed the Shortest Path Map (SPM) algorithm to identify key conformational hotspots in enzymes like retro-aldolase and monoamine oxidase (Casadevall et al., 2024; Romero-Rivera et al., 2017). Kamerlin's lab also utilized MD to study enzyme dynamics, particularly in $(\beta\alpha)_8$ barrel enzymes (Romero-Rivera et al., 2022). Another approach to flexibility design involves creating chimeras, where regions of one protein are transplanted into another. Notable examples include chimeric TEM1/PSE4 $\beta$-lactamases studied

with NMR (Morin et al., 2010) and Kamerlin's transplantation of the WPD loop in phosphatases (Moise et al., 2018; Shen et al., 2022). Damborsky's lab has focused on transplanting dynamic loops to explore flexibility and catalytic mechanisms in luciferases (Schenkmayerova et al., 2023; Chaloupkova et al., 2019), leading to the development of LoopGrafter, a web tool for *in silico* protein flexibility design (Planas-Iglesias et al., 2022).

**Protein inverse folding.** The task of predicting a sequence that folds into a given backbone structure (protein inverse folding) has become a key area where machine learning significantly contributes to protein design, along with structure prediction. ProteinMPNN (Dauparas et al., 2022), based on the graph neural network architecture introduced by Ingraham et al. (2019), is one of the most widely used tools for this task. Recent benchmarks, such as ProteinInvBench (Gao et al., 2023c), feature newer methods, e.g., KWDesign and PiFold (Gao et al., 2023b;a), which outperform ProteinMPNN in standard metrics such as sequence recovery on datasets like CATH4.3 (Pearl et al., 2003). Despite these advancements, ProteinMPNN remains the community standard, largely due to its extensive experimental validation. This highlights the limitations of such metrics as sequence recovery, which do not fully capture a model's ability to design functional or stable proteins (Wang et al., 2023). As a result, there is growing interest in developing more comprehensive evaluation metrics and tasks that better reflect real-world protein design challenges.

## 3 ANALYSIS OF DATA SOURCES FOR PROTEIN FLEXIBILITY

One of the key challenges in the engineering of protein flexibility is the lack of consistent and affordable methods for flexibility measurement or estimation. This hinders the development of flexibility datasets useful for learning. The most reliable type of data come from experiments and from physics-based simulations, in particular molecular dynamics simulations. However, experimental data are costly, time-consuming, and highly heterogeneous due to biases and different calibrations of the experiments. Similarly, molecular dynamics data are time-consuming and subject to many human-made choices, such as the choice of force field. Our goal in this section is to analyze the available data and data generation methods for protein flexibility that could be utilized for learning a fast flexibility predictor. We first specify how we quantify protein flexibility using relevant experimental and computational methods (Section 3.1). Then we critically compare these methods to identify a relevant data source for learning (Section 3.2).

### 3.1 METHODS FOR QUANTIFICATION OF PROTEIN FLEXIBILITY

In this work we focus on flexibility as a quantity describing the ability of individual residues (mainly their backbone) to move within the protein structure. To this end, we introduce in this section the relevant methods for quantification of protein flexibility, and in particular, the metrics used with each method for reporting per-residue protein flexibility values.

**MD.** Molecular Dynamics (MD) simulations can be used to quantify protein flexibility by simulating the atomic movements in a given modeled solvent environment. MD simulations are considered an accurate physics-based method and we thus regard them as the gold standard for flexibility estimation. However, their limitations should still be noted: (i) the simulation of the solvent is only approximate, affecting mainly the simulation accuracy of side-chain atoms, and (ii) potential interactions driven by molecules commonly present in the natural environment but not explicitly present in the simulation might affect the flexibility. For working with data on MD simulation, we select the recently published ATLAS dataset (Vander Meersche et al., 2023), particularly due to its excellent level of data curation. ATLAS consists of MD trajectories of 1390 proteins, each simulated in three replicas for 100ns per replica. For each protein and its residues, we quantify their flexibility by computing their root mean square fluctuations (RMSF) averaged over the three simulation replicas (see Appendix B for details on RMSF).

**B-factors.** The crystallographic B-factor carries information on protein flexibility, as it describes the inherent thermal vibrations of atoms in a crystal lattice. In such an organized environment, however, the natural movement of the protein is diminished, and crystal packing artifacts may lead to an underestimation of the real protein motions (Eyal et al., 2005). In this paper, we report B-factors obtained from the Protein Data Bank (PDB) (Berman et al., 2000). Note that for a few cases from PDB, we also kept the B-factors originating from NMR experiments, whose nature is different from those obtained in crystallographic experiments (for details, see Appendix A).

Table 1: Pearson correlation coefficients of flexibility predictions obtained by PDB B-factors and by computational methods (see Section 3.1 for details of the studied flexibility quantification methods). The reported coefficients are averaged over the 1383/1390 proteins from the ATLAS dataset (some were skipped due to missing pieces of structure resulting in NaNs from ENMs). As the table is symmetric, we show only values in the upper triangle. The best result in terms of correlation to MD simulations is highlighted in bold, and the second best is underlined.

| | MD | B-factors | AlphaFold2 | ESMFold | GNM | ANM |
|---|---|---|---|---|---|---|
| **MD** | - | 0.51 | 0.71 | 0.57 | 0.76 | **0.77** |
| **B-factors** | | - | 0.52 | 0.38 | 0.55 | 0.51 |
| **AlphaFold2** | | | - | 0.66 | 0.55 | 0.56 |
| **ESMFold** | | | | - | 0.45 | 0.43 |
| **GNM** | | | | | - | 0.94 |
| **ANM** | | | | | | - |

**AlphaFold2.** The protein structure prediction model AlphaFold2 (AF2) (Jumper et al., 2021) contains a confidence model predicting pLDDT scores. While the pLDDT score is primarily a measure of estimated confidence by AlphaFold2 (that may be induced by sequence variability in the region, for instance), it has been shown to be negatively correlated with flexibility (Ruff & Pappu, 2021; Guo et al., 2022; Saldaño et al., 2022) and exploited to approximate flexibility as measured by Nuclear Magnetic Resonance (NMR) (Ma et al., 2023). Therefore, to quantify flexibility with AF2, we use (1- pLDDT). Since the pLDDT prediction cannot be easily decoupled from the structure prediction, its runtime is the same as for the structure prediction, resulting in a relatively slow speed.

**ESMFold.** The protein structure prediction model ESMFold (Lin et al., 2022) also provides a pLDDT confidence model but with a significantly faster runtime than AlphaFold2. As in the case of AlphaFold2, we use (1-pLDDT) to quantify flexibility with ESMFold.

**GNM.** Gaussian Network Models (GNM) are a type of Elastic Network Models that model the atomic motions as isotropic, i.e., they model only the magnitudes of the displacements. To quantify flexibility, we use GNM to estimate the RMSF of each residue using the input structure from the ATLAS dataset that underwent a short MD relaxation. For details on the computation of RMSF for GNMs, see Appendix B.

**ANM.** In contrast, Anisotropic Network Models (ANM) model the atomic motions as anisotropic, i.e. they model also the direction of the displacement. As in the case of GNM, we quantify flexibility using RMSF estimated from an MD relaxed structure. For details on the computation of RMSF for ANMs, see Appendix B.

## 3.2 COMPARISON OF PROTEIN FLEXIBILITY QUANTIFICATION METHODS

Next, we compare the selected methods for quantification of protein flexibility on the ATLAS dataset (see Table 1). This is done via computing the correlation between the per residue flexibility profiles obtained through the different methods. The results of this analysis are discussed next.

Table 1 shows that the B-factors obtained for the ATLAS proteins from the PDB database do not correlate well with the other methods. We hypothesize that this is due to the crystal packing effect. The root mean square fluctuations (RMSF) from MD simulations thus appear to be a more reliable proxy for the true flexibility of single proteins (i.e., not in a crystal). We further support this conclusion by training a predictor of the B-factor data, which reached only modest performance, hinting at B-factor data being of insufficient quality for learning (see Appendix D). Hence, in the rest of the paper, we regard the flexibility computed using RMSF from the MD simulations (denoted further as MD) as the gold standard, which we will try to approach using learning.

Measured by the correlation to the RMSFs from the MD simulations, the best-performing computational methods are the Anisotropic Network Models (ANM), closely followed by the Gaussian Network Models (GNM) (see highlighted row in Table 1). Here, both ANM and GNM are evaluated over the MD relaxed structures from the ATLAS dataset and both result in mutually highly correlated RMSFs (Pearson correlation coefficient 0.94).

Having each simulation in three replicas in the ATLAS dataset allows us to evaluate the level of reliability of the RMSF computed using the short ATLAS MD trajectories. To this end, we compute

the correlation of RMSF between different simulation replicas of the same protein. We find that the different replicas mutually correlate on average with the Pearson correlation coefficient (PCC) of 0.88. We consider this number to be the indicative upper bound for the average correlation of any flexibility predictor trained and evaluated on the ATLAS dataset.

ANMs require a protein (backbone) structure on the input, which can be a limiting factor if such a structure is not available. While the ATLAS dataset provides MD relaxed structures, we also want to know how the ANMs perform outside of the ATLAS setting, for example, on structures coming from (i) the PDB, (ii) protein folding of the original sequence by ESMFold, or (iii) protein folding of an alternative sequence obtained with ProteinMPNN (Dauparas et al., 2022). Therefore, we investigate the effect of the type of input structure on the performance of ANMs in Table 6 in Appendix E, where we show that redesigning the sequence of a protein with ProteinMPNN does not have a significant effect on the performance of ANM.

## 4    METHODS

In this section, we introduce our new methods for predicting protein flexibility (Section 4.1) and the new Flexpert-Design framework demonstrating the applicability for the problem of engineering protein flexibility (Section 4.2).

### 4.1    LEARNING TO PREDICT PROTEIN FLEXIBILITY

Our above analysis of methods for protein flexibility quantification (see Section 3.2) motivates the effort to develop a new protein flexibility predictor as there is still enough room for improvement in terms of correlation to MD (compare the PCC of 0.77 for ANM with the indicative upper bound on PCC of 0.88). Here we address the challenge by introducing a novel and fast predictor of protein flexibility in two versions: Flexpert-Seq, taking a protein sequence on the input, and Flexpert-3D, taking both sequence and structure on the input. In both versions, the method learns to predict the RMSF of MD simulations in a supervised manner using the ATLAS dataset. The two predictors are described next.

**Flexpert-Seq: Predicting flexibility from sequence.**    The goal is to develop a predictor of protein flexibility that uses only a protein sequence on the input and learns to predict the protein sequence flexibility as close as possible to the predictions from MD trajectories. In detail, given a dataset $\mathcal{D}_{Seq} = \{(\boldsymbol{s_i}, \boldsymbol{y_i})\}_{i \in D}$ of protein sequences $\boldsymbol{s_i} \in \mathcal{A}^{N_i}$ and per residue flexibility labels $\boldsymbol{y_i} \in \mathbb{R}^{N_i}$, where $D$ is the set indexing the protein in the dataset, $\mathcal{A}$ is the alphabet of 20 natural amino acids and $N_i$ is the number of amino acids of protein $i$, the goal is to train a regression model $\mathcal{F}_{Seq} : \mathcal{A}^{N_i} \to \mathbb{R}^{N_i}$ to predict the per residue flexibility from a protein sequence. To overcome the limitations coming from the small amount of annotated data (1390 proteins in the ATLAS dataset), we leverage the pretrained protein language model Prot-Trans (Elnaggar et al., 2022) to obtain

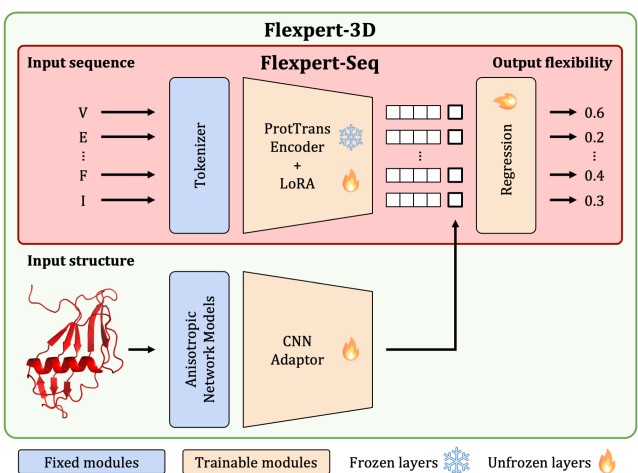

Figure 1: Overview of the architecture of our protein flexibility prediction methods Flexpert-Seq (the red box) and Flexpert-3D (the whole architecture). Icons of snowflake and flame denote frozen and unfrozen layers, respectively, in modules containing trainable parameters. Trainable (orange) and fixed (blue) modules denote whether any trainable unfrozen parameters are present in the module.

per-residue embeddings, on top of which we learn a linear layer to regress a single scalar value for each residue. An alternative choice for the backbone protein language model could be the ESM-2 model, which was shown to perform similarly to ProtTrans in fine-tuning for diverse downstream

tasks (Schmirler et al., 2024). Following the procedure proposed by Schmirler et al. (2024), we employ LoRA (Hu et al., 2021a) to fine-tune ProtTrans together with training the regression layer for our downstream task. See Figure 1 for an overview of the method.

**Flexpert-3D: Predicting flexibility from sequence and structure.** Since inverse folding pipelines usually have access to protein backbones on the input, we also explore the strategy of using both protein sequence and protein backbone structure on the input to predict the protein sequence flexibility as close as possible to the predictions from MD trajectories. More formally, given a dataset $\mathcal{D}_{3D} = \{(s_i, \chi_i, y_i)\}_{i \in D}$ of protein sequences $s_i \in \mathcal{A}^{N_i}$, protein backbone structures $\chi_i \in \mathbb{R}^{4 \times 3N_i}$ (4 atoms per residue backbone) and per residue flexibility labels $y_i \in \mathbb{R}^{N_i}$, where $D$ is the set indexing the protein in the dataset, $\mathcal{A}$ is the alphabet of 20 natural amino acids and $N_i$ is the number of amino acids of protein $i$, the goal is to train a regression model $\mathcal{F}_{3D} : \mathcal{A}^{N_i} \times \mathbb{R}^{4 \times 3N_i} \rightarrow \mathbb{R}^{N_i}$ to predict the per residue flexibility from protein sequence and backbone structure. As ANMs can be run relatively fast if the protein backbone is available and they were the best performing method as evaluated in Section 3.2, we tackle the task by combining our sequence-based method Flexpert-Seq with ANM-based flexibility values. This strategy is thus akin to learning how to correct the crude flexibility annotations returned by ANMs to match more accurate values derived from MD trajectories. To this end, we enhance the architecture of Flexpert-Seq by a CNN head acting as an adaptor of the ANM-predicted flexibility into the embedding space of Flexpert-Seq (see Figure 1). We train the CNN Adaptor together with the LoRA layers by fine-tuning the ProtTrans Encoder of Flexpert-Seq and, together with the regression layer, mapping the combined embeddings to per-residue flexibility.

## 4.2 Flexpert-Design: Flexibility engineering via flexibility-aware inverse folding

In this section, we apply our flexibility predictor Flexpert-3D for the task of engineering protein flexibility. The goal is to generate a protein sequence that respects a given protein backbone structure and a set of flexibility instructions.

More formally, we are given a dataset $\mathcal{D}_{bb} = \{(s_i, \chi_i)\}_{i \in D}$ of protein sequences $s_i \in \mathcal{A}^{N_i}$ and protein backbone atom coordinates $\chi_i \in \mathbb{R}^{4 \times 3N_i}$ (4 atoms per residue backbone), where $D$ is the set indexing the protein in the dataset, $\mathcal{A}$ is the alphabet of 20 natural amino acids and $N_i$ is the number of amino acids of protein $i$. We are also given a set of flexibility instructions $F = \{f_i\}_{i \in D}$, where $f_i \in \mathbb{R}^{N_i}$. The goal is to train model $\mathcal{P}_F : \mathbb{R}^N \times \mathbb{R}^{4 \times 3N} \rightarrow \mathcal{A}^N$, which takes $N$ flexibility instructions and a backbone structure of a protein of length $N$ and maps it to a protein sequence of the length $N$, respecting the input backbone and the flexibility instructions.

We tackle the problem by leveraging the established protein inverse folding model ProteinMPNN and modifying it to take the flexibility instruction on the input. We then devise a learning strategy where we utilize our flexibility predictor Flexpert-3D to fine-tune the inverse folding model toward respecting the flexibility instructions. Details are given next.

**Teaching inverse folding model to engineer flexibility.** The critical aspects of our flexibility engineering task are the choice of the flexibility engineering instructions $F$ and the design of the learning strategy steering the inverse folding model to follow the instructions.

Our novel training strategy is to inform the inverse folding model about the flexibility of the native sequence and to optimize the model toward generating sequences with flexibility preserved with respect to the input. We hypothesize that such training makes the model pay attention to the flexibility instructions $f \in F$ provided on the input. During inference, this awareness enables us to exploit the flexibility input of the model to pass our actual instructions for modifying the flexibility with respect to the native one.

For learning the flexibility of native sequences, we propose the following sequence of steps. First, we construct the set $F^{native}$ of flexibility pseudolabels obtained with Flexpert-3D as

$$F^{native} = \{f_i^{native} | f_i^{native} = \mathcal{F}_{3D}(s_i, \chi_i), (s_i, \chi_i) \in D_{bb}, i \in D\}, \tag{1}$$

where the Flexpert-3D predictor is used to predict flexibility for all tuples of sequences $s_i$ and backbone coordinates $\chi_i$ from dataset $\mathcal{D}_{bb}$. These predicted flexibilities $f_i^{native}$, which we call "native", are used in training to inform the inverse folding model about flexibility.

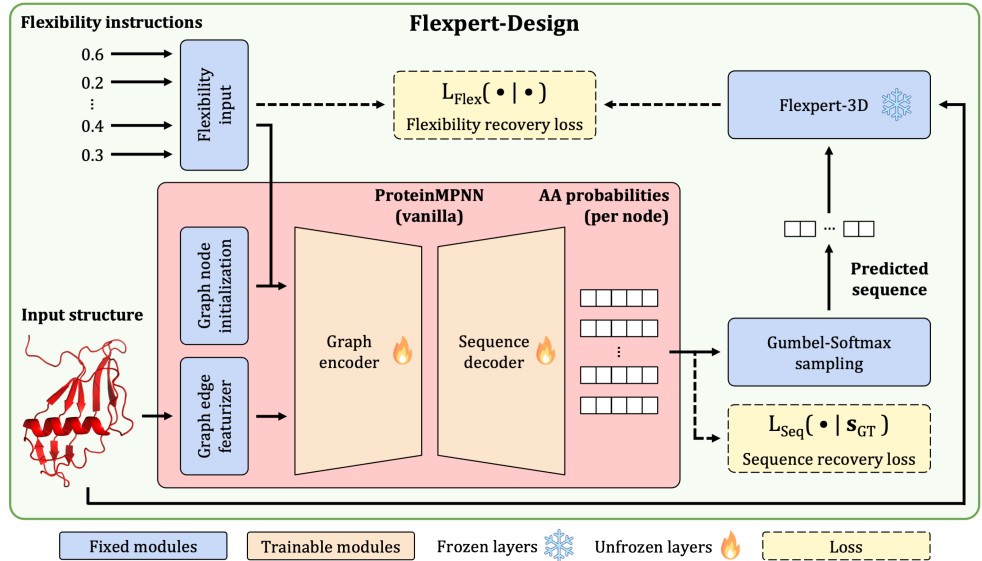

Figure 2: Overview of the Flexpert-Design method for fine-tuning of inverse folding models toward steerability by requirements on protein flexibility. The inverse folding model ProteinMPNN, taking protein backbone structure on the input, is modified to additionally take on the input flexibility instructions. This is implemented by adding a flexibility instruction to the zero-initialized node feature for each residue. The predicted sequence is sampled in a differentiable way using Gumbel-Softmax and passed to Flexpert-3D together with the input structure. Note that we are making use of the assumption that ProteinMPNN does not modify the backbone structure as we are matching together the newly predicted sequence with the ground truth backbone structure. The flexibility of the sequence predicted by Flexpert-3D is passed to the loss function $L_{Flex}$ which computes the distance between the input flexibility instructions and the flexibility of the predicted sequence. Icons of snowflake and flame denote frozen and unfrozen layers, respectively, in modules containing trainable parameters. Trainable (orange) and fixed (blue) modules denote whether any trainable unfrozen parameters are present in the module.

Second, we take a vanilla ProteinMPNN inverse folding model $\mathcal{P} : \mathbb{R}^{4 \times 3N} \rightarrow \mathcal{A}^N$, which maps protein backbones of length $N$ to protein sequences of the same length, and modify it to a flexibility-aware ProteinMPNN model $\mathcal{P}_F : \mathbb{R}^N \times \mathbb{R}^{4 \times 3N} \rightarrow \mathcal{A}^N$, which accepts flexibility instructions $\boldsymbol{f} \in F \subset \mathbb{R}^N$ on the input as well (see Figure 2).

Third, we introduce a new loss function $\mathcal{L}_{Flex}$ for guidance of the inverse folding model $\mathcal{P}_F$ toward preservation of the input flexibility by matching the input flexibility instructions $\boldsymbol{f}$ to the output flexibility $\mathcal{F}_{3D}(\hat{\boldsymbol{s}}, \boldsymbol{\chi})$:

$$\mathcal{L}_{Flex} = \mathcal{L}(\mathcal{F}_{3D}(\hat{\boldsymbol{s}}, \boldsymbol{\chi}), \boldsymbol{f}), \tag{2}$$

where $\mathcal{L}$ is a regression loss (MSE or L1), $\boldsymbol{\chi}$ is the input backbone structure, and $\hat{\boldsymbol{s}} = \mathcal{P}_F(\boldsymbol{f}, \boldsymbol{\chi})$ is the sequence predicted by the flexibility-aware inverse folding model $\mathcal{P}_F$. We illustrate our method, including the loss function $\mathcal{L}_{Flex}$, in Figure 2. Using the pseudolabels of flexibility $F^{native}$ obtained by Flexpert-3D as the flexibility instructions in training, the $L_{Flex}$ loss ensures in the training process that the model learns to preserve flexibility it has seen on the input. During inference, the learned flexibility preservation is exploited to pass non-native flexibility instructions to the model to steer the inverse folding toward generating sequences with the required flexibility.

**Training to recover flexibility and sequence.** To train ProteinMPNN toward flexibility awareness, we still need to ensure it also works for its original objective of generating a protein sequence that fits the input backbone. Therefore, we first train ProteinMPNN using its standard Cross-Entropy loss $\mathcal{L}_{Seq}$ and optimize it for maximal sequence recovery. Taking the pre-trained ProteinMPNN model weights, we further fine-tune them using the combined $\mathcal{L}_{Flexpert}$ loss:

$$\mathcal{L}_{Flexpert} = \theta \cdot \mathcal{L}_{Flex} + (1 - \theta) \cdot \mathcal{L}_{Seq}, \tag{3}$$

where $\theta \in [0, 1]$ is the parameter mixing the two losses $\mathcal{L}_{Flex}$ and $\mathcal{L}_{Seq}$ into a single loss. The choice of $\theta$ and the sequence recovery of the trained models is discussed in Appendix H.

Table 2: Pearson correlation coefficients between fast sequence-based baselines (AlphaFold2 and ESMFold) and our method Flexpert-Seq (columns) and the RMSF values from MD simulations and from Anisotropic Network Models (ANM) evaluated over MD relaxed structures (rows). Note that an upper bound on correlation to MD is estimated to be 0.88. Evaluation performed for 139 proteins from ATLAS testset. Our approach outperforms the AlphaFold2 and ESMFold baselines and obtains the best correlation with the flexibility estimates obtained from Molecular Dynamics (see the value in bold). Interestingly, the correlation coefficient of our approach and Anisotropic Network Models (ANM) is just 0.66, suggesting that both our approach and ANM might be complementary, motivating the development of a predictor combining the sequence-based predictor with ANM.

|  | AF2 | ESMFold | Flexpert-Seq (ours) |
| --- | --- | --- | --- |
| **MD** | 0.72 | 0.58 | **0.78** |
| **ANM** | 0.56 | 0.43 | 0.66 |

**Preserving differentiability through discrete language model input.** An interesting challenge in utilizing the protein language model-based predictor Flexpert-3D inside of an end-to-end training pipeline is the question of preserving differentiability at the interface of the inverse folding model output (continuous probability distribution) and the language model input (discrete tokens) (see Figure 2). Our solution to this problem is to sample the learned distribution of amino acids using the Gumbel-Softmax trick (Jang et al., 2017). In particular, we use its straight-through variant, which takes the argmax of the input distribution in the forward pass but preserves the gradients by approximating the argmax with the Gumbel-Softmax distribution in the backward pass. This enables us to sample a discrete sequence for the language model while preserving the gradients and enabling end-to-end training of the entire pipeline.

## 5 RESULTS

In this section, we report the performance of our methods Flexpert-Seq and Flexpert-3D in terms of protein flexibility prediction (Section 5.1) and the performance of Flexpert-Design in terms of increasing flexibility in engineered protein regions (Section 5.2).

### 5.1 FLEXIBILITY PREDICTION

**Experimental setup.** We use the sequences and MD relaxed structures from the ATLAS dataset as our inputs. The flexibility labels are the per residue RMSFs averaged over the 3 MD replicas for each protein. We split the dataset using topology splitting, with the additional requirement that no topologies present in the CATH4.3 test set are present in our ATLAS training set. The reason is to prevent potential leakage when combining the flexibility predictors into the Flexpert-Design pipeline which is trained using CATH4.3. We evaluate the Pearson correlation coefficient (PCC) between the predicted flexibility and the ground truth flexibility (RMSF) for each protein in the ATLAS test set. We report the average PCC across the test set.

**Results.** Our predictors Flexpert-Seq and Flexpert-3D outperform the baselines when using only the sequence on the input and when also using the structure, respectively. See Table 2 for the sequence case and Table 3 for the case when using sequence and structure. Note that our predictor Flexpert-3D, which partly relies on ANM, improves over them significantly and does not rely on them exclusively (see PCC of 0.78 to ANM).

We further tested our predictors Flexpert-Seq and Flexpert-3D on a separate dataset called mdCATH (Mirarchi et al., 2024), which features MD trajectories obtained at various simulation temperatures ranging from 320 K to 450 K. Using this data, we test the effect of the simulation temperature on the performance of our models trained on the ATLAS dataset simulated at a physiological temperature of 300 K. The experiment confirmed that the further the temperature is from the simulation temperature in the training set, the lower the Pearson correlation coefficients are between the model predictions and the RMSFs, see Appendix F.

In Appendix G, we provide qualitative examples of Flexpert-Seq and Flexpert-3D predictions and how these predictions compare to the ground truth RMSF and to the predictions of ANM.

Table 3: Pearson correlation coefficients between fast structure-based baselines (Anisotropic Network Models and Gaussian Network Models) and our method Flexpert-3D (columns) and the RMSF values from MD simulations and from Anisotropic Network Models (ANM) evaluated over MD relaxed structures (rows). Evaluation is performed for 139 proteins from ATLAS test set. Note that an upper bound on correlation to MD is estimated to be 0.88. Also, note that our approach outperforms the ANM and GNM baselines and obtains the best correlation with the flexibility estimates obtained from Molecular Dynamics (row MD). The correlation coefficient of our model with ANM is 0.78 (see the bottom row), indicating that our model relies on ANM input only partially and actually learns novel information from the sequence.

|  | ANM | GNM | Flexpert-3D (ours) |
|---|---|---|---|
| **MD** | 0.76 | 0.76 | **0.83** |
| **ANM** | 1.00 | 0.93 | 0.78 |

## 5.2 PROTEINMPNN CAN BE STEERED BY FLEXIBILITY INSTRUCTIONS

**Experimental setup.** The evaluation is done using the CATH4.3 dataset. The set of flexibility-increasing instructions $F_\uparrow^{\tau,S}$ is constructed in the following way: (i) The flexibility instructions are initialized by the native flexibilities $F^{native}$ (as defined in Equation (1)); (ii) For each protein independently, a random contiguous segment $S$ of length $|S|$ is subselected from the sequence. In case $|S| \geq N$, where $N$ is the length of the protein, the whole protein is selected as the segment; (iii) For each residue $i$ in each selected segment $S$, its flexibility instruction $f_i$ is incremented by $\tau > 0$, aiming at increasing the residues flexibility:

$$f_i = f_i^{native} + \tau \cdot \mathbb{1}_{i \in S}, \tag{4}$$

where $\mathbb{1}_{i \in S}$ is the indicator function for the set of amino acids corresponding to the segment $S$.

We then take the set of flexibility increasing instructions $F_\uparrow^{\tau,S}$ together with the corresponding backbones $\mathcal{D}_{bb}$ from the CATH4.3 test set and pass it to the flexibility-aware ProteinMPNN $\mathcal{P}_F$ trained using our Flexpert-Design method to obtain predicted protein sequences $\hat{s}$:

$$\hat{s}_j = \mathcal{P}_F(\boldsymbol{f}_j, \boldsymbol{\chi}_j), \forall (\boldsymbol{f}_j, \boldsymbol{\chi}_j) \in F_\uparrow^{\tau,S} \times \mathcal{D}_{bb}. \tag{5}$$

We evaluate the predicted sequences using the metrics described next. For each predicted sequence $\hat{s}_j$, we identify the mutated residues $\mathcal{M}_j \subset S_j$ inside the segment $S_j$ engineered for increased flexibility. For such residues, we define the flexibility enrichment ratio $r_{ij}$:

$$r_{ij} = \frac{\mathcal{F}_{3D}(\hat{s}_j, \boldsymbol{\chi}_j)_i}{f_{ij}^{native}}, \tag{6}$$

where $\hat{s}_j$ is the predicted sequence for protein $j$ with backbone $\boldsymbol{\chi}_j$, $i \in \mathcal{M}_j$ indexes the mutated residues in the engineered region $S_j$, and $f_{ij}^{native}$ is the native flexibility of the residue (see Equation (1)). The flexibility enrichment ratios $r_{ij}$ are then used to report the median enrichment ratio and the proportion of flexibility-increasing mutations, i.e., those for which $r_{ij} > 1$.

**Results.** Table 4 shows that our method Flexpert-Design successfully managed to engineer increased flexibility in the engineered region. This result is presented in more detail in Figure 3, which shows that Flexpert-Design can shift the flexibility distribution in the engineered segment toward higher flexibility. Furthermore, Figure 3 shows the change in the distribution of amino acid types in the engineered region. The observation that the proportion of Alanine (A) and Glycine (G) increases goes in line with the biochemical understanding that these amino acids are smaller and tend to be more flexible. The sensitivity of Flexpert-Design to the size of the engineered region $|S|$ and the value of the flexibility increasing instruction $\tau$ is studied in Appendix J.

Interestingly, the most contributing component of Flexpert-Design is the inclusion of flexibility input and training with the Flexpert-3D generated flexibility pseudolabels. The additional optimization using $\mathcal{L}_{Flex}$ brings only relatively modest improvements in steerability (see Table 4 and Figure 8). Please compare "Flexpert-Design (without loss)", which does not use $\mathcal{L}_{Flex}$, and the full model denoted "Flexpert-Design". The results also clearly demonstrate significant improvements of our

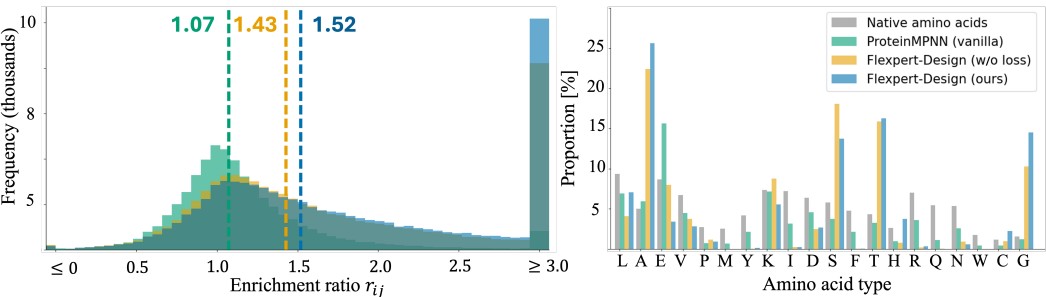

Figure 3: Comparison of the enrichment ratio distributions (left) and the amino acid type distributions (right) for the flexibility-increasing experiment with segment lengths $|S| = 50$ and $\tau = 5$. Our Flexpert-Design model is compared to its variant that was trained without the $\mathcal{L}_{Flex}$ loss but with the flexibility on the input. Vanilla ProteinMPNN is presented as a baseline. On the left, the median values for each distribution are denoted with a vertical line. On the right, the native amino acid distributions in the examined segments are presented for reference.

Table 4: Comparison of the ability of flexibility engineering models to increase flexibility of engineered segments. The presented median enrichment ratios $\text{Med}(r_{ij})$ and the proportion of flexibility-increasing mutations were obtained using the CATH4.3 test set with engineered segments of length $|S| = 50$, and flexibility engineering instructions corresponding to native flexibility incremented by $\tau = 5$ in the engineered segment. The best results in each category are highlighted in bold, second best are underlined.

|  | Median enrich. ratio $\text{Med}(r_{ij})$ ↑ | Prop. of flexibility incr. mutations ↑ |
| --- | --- | --- |
| **Flexpert-Design (ours)** | **1.52** | **0.83** |
| **Flexpert-Design (w/o loss)** | 1.43 | 0.80 |
| **ProteinMPNN (baseline)** | 1.07 | 0.61 |

model in the ability to engineer flexibility compared to the ProteinMPNN baseline. Additional results in Section H of the Appendix also demonstrate virtually no loss in sequence recovery metrics of our approach compared to the ProteinMPNN baseline. The analysis of the structure preservation of the sequences with engineered increased flexibility is given in Appendix I.

We also investigated the use of our model for engineering of decrease in flexibility, results of which we discuss in Appendix K.

## 6 CONCLUSIONS

In this work, we address the problem of engineering protein flexibility—a highly relevant task due to its potential impact on the field of protein design. Firstly, we analyze the available data obtained in wet-lab experiments, physics-based simulations, or as a byproduct of machine learning-based structure prediction models. We identify the flexibility estimate by molecular dynamics simulation as the best learning target, set an upper bound on the potential performance of the learned predictor, and analyze the effect of the type of input protein structure on the performance of Anisotropic Network Models. Secondly, we develop a protein language model-based predictor of protein flexibility working either with protein sequence (Flexpert-Seq) or with both sequence and protein backbone structure on the input (Flexpert-3D) and show that our predictors outperform the relevant baselines. Thirdly, we propose a learning strategy Flexpert-Design, which uses the Flexpert-3D flexibility predictor to steer an inverse folding model toward a generation of sequences with increased flexibility. We demonstrate that inverse folding models can be steered toward generating protein sequences with increased flexibility as measured by our protein flexibility predictor. This opens up new possibilities for development of proteins with enhanced biological activities. In some practical protein engineering tasks, it would also be of interest to engineer protein sequences with decreased flexibility, where the ability of our current Flexpert-Design model seems limited (see Appendix K). This motivates future development of new models, which would provide a tighter control of the designed sequence's flexibility and which would be successful regardless of whether the flexibility was instructed to increase or decrease. An interesting future direction could be to use Flexpert-3D or Flexpert-Seq for guidance of discrete flow matching models.

REPRODUCIBILITY STATEMENT

The code, together with the instructions on how to download the data and trained weights for the reproduction of this work, can be found at `https://github.com/KoubaPetr/Flexpert`.

ACKNOWLEDGEMENTS

This work was supported by the Ministry of Education, Youth and Sports of the Czech Republic through projects ELIXIR (No. LM2023055), ESFRI RECETOX RI (No. LM2023069), and OP JAK CLARA (No. 02_23_029). We acknowledge VSB – Technical University of Ostrava, IT4Innovations National Supercomputing Center, Czech Republic, for awarding this project access to the LUMI supercomputer, owned by the EuroHPC Joint Undertaking, hosted by CSC (Finland) and the LUMI consortium through the Ministry of Education, Youth and Sports of the Czech Republic through the e-INFRA CZ (No. 90254). This work was also supported by the European Union (ERC project FRONTIER (No. 101097822), ELIAS (No. 101120237), CLARA (No. 101136607)), the Technology Agency of the Czech Republic under the NCC Programme from the state budget (No. RETEMED TN02000122) and under the NRP from the EU RRF (No. TEREP TN02000122/001N), and the CETOCOEN EXCELLENCE (No. 857560) projects supported from the European Union's Horizon 2020 research and innovation programme. Views and opinions expressed are however those of the author(s) only and do not necessarily reflect those of the European Union or the European Research Council. Neither the European Union nor the granting authority can be held responsible for them. Petr Kouba is a holder of the Brno Ph.D. Talent scholarship funded by the Brno City Municipality and the JCMM.

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

APPENDIX

## A  EXPERIMENTAL MEASUREMENT OF PROTEIN FLEXIBILITY BY NMR AND HDX-MS

In contrast to crystallographic methods directly obtaining the B-factors (temperature factors), NMR reads chemical shift perturbations, in other words, how the proximity of different atoms affects the chemical environment of each other. As a product of these measurements, atom-to-atom distance restraints can be inferred. Protein flexibility can be derived as the change in the intensity and amplitude of the read signals over time; thus, different sets of restraints can be inferred for different time points (Hu et al., 2021b). Hence, NMR usually provides an ensemble of structural models, from which atomistic Root Mean Square Fluctuations (RMSFs) can be derived, which can be converted to B-factors by the following equation:

$$B_i = \frac{8\pi^2}{3}(RMSF_i)^2,\tag{7}$$

where $B$ is the B-factors and $i$ represents an individual atom (Emperador et al., 2010).

The HDX-MS relies on the exchange of isotopic hydrogen in a deuterium-rich medium over a given span of time to detect the protein segments that are more exposed to solvent. Flexible regions (which often lay on the protein surface away from the hydrophobic tightly packed protein core) allow for higher solvent penetrance and, thus, attain higher deuteration values when compared to less flexible areas (Uhrik et al., 2023).

## B  ROOT MEAN SQUARE FLUCTUATIONS (RMSF) AS A MEASURE OF FLEXIBILITY

The Root Mean Square Fluctuations (RMSF) are a way of quantifying protein flexibility, typically based on simulated data of dynamics or based on simplified models of dynamics such as the Elastic Network Models (ENM). In this work, we use RMSF computed on top of both MD simulations and ENM (Anisotropic Network Models and Gaussian Network Models). These RMSFs were obtained in the following way:

**RMSF from MD simulations.**  We used the values provided directly with the ATLAS dataset, which provides the per residue RMSF for each of its simulation replicas. To represent each protein's flexibility profile, we used the per residue RMSF averaged over all 3 simulation replicas of each protein. The values provided by the ATLAS dataset were computed as standard deviation of the $\alpha$-carbon displacements, using the GROMACS software. That is, the RMSF for MD simulations were computed using the following formula:

$$\rho_i = \sqrt{\langle ||r_i - \langle r_i \rangle||_2^2 \rangle},\tag{8}$$

where $\rho_i$ denotes the RMSF of residue $i$, $r_i$ stands for the coordinates of the $\alpha$-carbon of residue $i$ in the global reference frame, the angle brackets $\langle \ \rangle$ stand for the average over all the frames of the simulation and $|| \cdot ||_2$ denotes the Euclidean norm.

**RMSF for Elastic Network Models (Anisotropic Network Models).**  We used the ProDy (Bakan et al., 2011) package to compute the ENMs and the corresponding RMSF values. Firstly, we used ProDy to build the Hessian matrix, based on the $\alpha$-carbon coordinates of our input structure and the cutoff of 16 Angstrom, then the non-trivial modes of the Hessian were calculated, and consequently these modes were used to solve the problem of finding the inverse of the Hessian to compute the fluctuations according to the formula for the ENMs:

$$\rho_i = \sqrt{\frac{1}{C}\mathrm{Tr}([H^{-1}]_{ii})},\tag{9}$$

Table 5: Pearson correlation coefficients of MD flexibility and Flexpert-Seq predictions, when trained on B-factors (B) and per protein normalized B-factors ($B^{norm}$).

| | Flexpert-Seq (trained on B) | Flexpert-Seq (trained on $B^{norm}$) |
|---|---|---|
| **MD** | 0.47 | 0.56 |

where $\rho_i$ denotes the RMSF of residue $i$, $C$ denotes a constant that is typically equal to the product of the Boltzmann constant and the temperature ($C = k_B T$), but for us this is arbitrary as we are interested in the correlations of the fluctuations and the correlations are not affected by the scaling by a constant. The $\mathrm{Tr}$ in the formula stands for the trace of the matrix $[H^{-1}]_{ii}$ which is the 3x3 submatrix of the inverse Hessian corresponding to the residue $i$ (the 3x3 submatrix comes from the fact that the $\alpha$-carbon positions are represented in 3D).

**RMSF for Elastic Network Models (Gaussian Network Models).** For Gaussian Network models, the procedure is analogous to the case of Anisotropic Network Models. The only difference is that instead of Hessian matrix, the Kirchoff matrix is used:

$$\rho_i = \sqrt{\frac{1}{C}\mathrm{Tr}([\Gamma^{-1}]_{ii})}, \tag{10}$$

where $\Gamma$ is the Kirchhoff matrix constructed for the $\alpha$-carbon coordinates of the input structure considering the cutoff of 16 Angstrom.

More details on calculation of fluctuations from ENMs (both Anisotropic and Gaussian Network Models) can be found in (Yang et al., 2009).

## C  FINER-GRAINED ELASTIC NETWORK MODELS

The complexity of the ENM model can increase by considering different types of beads and springs (i.e., each amino acid and all the possible combinations of pairs) or making them finer-grained (i.e., considering more than one bead per amino acid), approaching the model at the atomistic resolution. The order of the algorithm increases at least quadratically with the complexity of the model, making these solutions more accurate but less efficient. Some implementations of modified ENMs are available in ENCoM (Frappier & Najmanovich, 2014) and DynaMut (Rodrigues et al., 2021).

## D  THE DIFFICULTY OF LEARNING FROM B-FACTORS COMPARED TO MD TRAJECTORIES

We also investigated the potential of B-factors to be used as data for learning to predict protein flexibility. To this end, we retrained Flexpert-Seq using B-factors instead of the RMSFs from MD trajectories. Because the absolute values of B-factors strongly depend on the calibration of the experiment, we trained both on the raw B-factors as well as on the B-factors standardized over each protein chain separately, using the following equation:

$$B_i^{norm} = \frac{B_i - \bar{B}}{\sigma(B)} \tag{11}$$

where $B_i$ is the raw B-factor of residue $i$, $\bar{B} = \frac{1}{N}\sum_{j=1}^{N} B_j$ is the average raw B-factor of a protein of size $N$ and $\sigma B = \sqrt{\frac{1}{N-1}\sum_{j=1}^{N}(B_j - \bar{B})^2}$ is the standard deviation of the raw B-factors across protein $i$.

Table 5 reports the performance of Flexpert-Seq trained to predict B-factors and normalized B-factors. We see the performance is significantly compromised compared to the results of Flexpert-Seq trained on MD data (Pearson correlation coefficient to MD of 0.76). We suspect that the experimental data suffer from the crystal packing effect, and they are too noisy to reliably learn from.

Table 6: Pearson correlation coefficients between root mean square fluctuations (RMSF) from molecular dynamics (MD) simulations (row) and flexibility predictions obtained by Anisotropic Network Models (ANM) over differently processed input structures (columns).

| | ANM (MD str.) | ANM (PDB str.) | ANM (ESMFold str.) | ANM (pMPNN + ESMFold str.) |
|---|---|---|---|---|
| **MD** | 0.77 | 0.73 | 0.73 | 0.71 |

## E   THE EFFECT OF TYPE OF INPUT STRUCTURE ON THE PERFORMANCE OF ANISOTROPIC NETWORK MODELS

Table 6 shows that the PDB structure obscures some of the flexibility, probably due to the crystal packing effect present in most cases. Similarly, the ANM performance is diminished when a structure obtained by ESMFold is given on the input. Interestingly, preceding the ESMFold by applying ProteinMPNN to generate an alternative sequence leads to only a small additional drop in performance. This is explained by the fact that the standard ANMs implemented in ProDy (Bakan et al., 2011), which we used, do not consider the amino-acid identities and only work with the coordinates of the backbone. And since ProteinMPNN is optimized to preserve the backbone geometry, it has only a marginal effect on the performance of ANMs.

## F   EVALUATION OF FLEXPERT-3D AND FLEXPERT-SEQ ON MDCATH DATASET

To evaluate the generalization capability of Flexpert-3D and Flexpert-Seq predictors outside of the ATLAS dataset, we performed an additional experiment evaluating the predictors on the mdCATH dataset (Mirarchi et al., 2024).

The recently introduced mdCATH dataset presents a challenging test of generalizability for our predictors trained on the ATLAS dataset. The mdCATH dataset is larger in terms of the number of data points (5398 compared to 1390 in ATLAS) and the simulated data correspond to individual protein domains (i.e., they are shorter proteins, on average with 137 residues compared to 236 residues in ATLAS). Furthermore, the simulation times in mdCATH are longer than in ATLAS (on average 464 ns for mdCATH compared to 100 ns for ATLAS) and the simulations were performed at different temperatures (320 K, 348 K, 379 K, 413 K, and 450 K) than the simulations in the ATLAS dataset (300 K). The results of the evaluation are presented in Table 7.

Table 7: Evaluation of Flexpert-3D and Flexpert-Seq predictors on MD simulations from the md-CATH dataset. Both predictors were evaluated on the full dataset (5398 proteins, columns "Full data") as well as on its subset excluding topologies present in the ATLAS training set (4013 proteins, columns "Topo. filtered"). The reported numbers are Pearson correlation coefficients between the model predictions and the RMSF from the dataset averaged over all the proteins in the dataset.

| | Flexpert-3D | Flexpert-Seq |
|---|---|---|
| **Simulation temperature** | **Full data / Topo. filtered** | **Full data / Topo. filtered** |
| $T = 320$ K | 0.69 / 0.66 | 0.64 / 0.64 |
| $T = 348$ K | 0.64 / 0.63 | 0.63 / 0.63 |
| $T = 379$ K | 0.59 / 0.57 | 0.60 / 0.59 |
| $T = 413$ K | 0.49 / 0.48 | 0.52 / 0.52 |
| $T = 450$ K | 0.34 / 0.33 | 0.38 / 0.38 |

Despite some performance drop with respect to the performance achieved on the ATLAS dataset (0.82 for Flexpert-3D and 0.76 for Flexpert-Seq), the performance of our predictors remains reasonable for the simulations performed at the temperature T = 320 K (0.66 for Flexpert-3D and 0.64 for Flexpert-Seq, for the topology filtered dataset). Considering that the temperature in this part of the mdCATH dataset is 20 K away from the temperature of the simulations in the ATLAS training

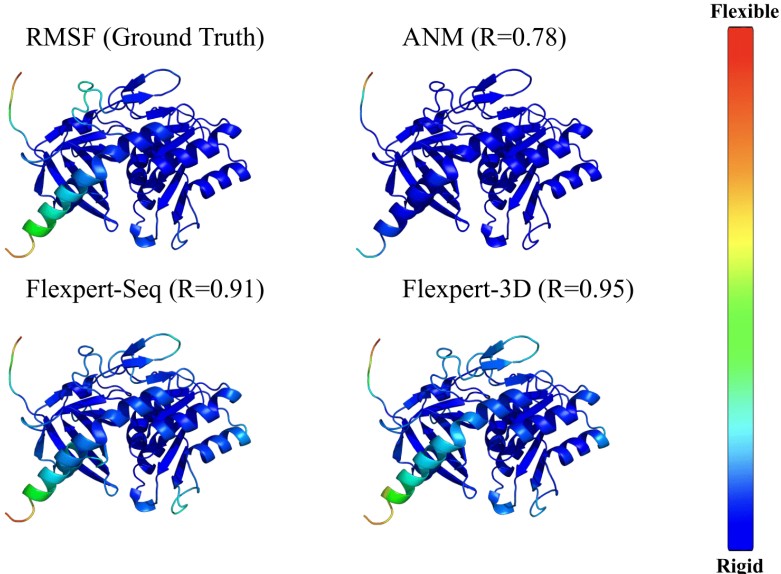

Figure 4: ATLAS datapoint 5jca_S. Flexpert-Seq and Flexpert-3D seem to improve over the prediction of ANMs mainly by correctly capturing the higher flexibility of the helix at the terminus.

set, this appears as a reasonable generalization. We also notice that Flexpert-3D experiences a more dramatic decrease in performance than Flexpert-Seq for higher temperatures in mdCATH simulations, suggesting that the sole reliance on sequence information makes the method more robust to the change in the simulation temperature. In addition, the proteins are possibly getting unfolded in the high-temperature simulations of mdCATH, leading to non-physiological degrees of flexibility.

## G    QUALITATIVE ANALYSIS OF FLEXPERT-SEQ AND FLEXPERT-3D FLEXIBILITY PREDICTIONS

In this section, we present some qualitative examples of Flexpert-Seq and Flexpert-3D predictions in the form of visualization of protein structures colored by the flexibility. The visualized flexibility is predicted by Flexpert-Seq and Flexpert-3D or estimated by MD (considered as the ground truth) or by Anisotropic Network Models (ANM). See Figures 4 to 7.

## H    BALANCING SEQUENCE RECOVERY AND FLEXIBILITY AWARENESS IN FLEXPERT-DESIGN TRAINING

When training Flexpert-Design using the combined loss $\mathcal{L}_{Flexpert} = \theta \cdot \mathcal{L}_{Flex} + (1-\theta) \cdot \mathcal{L}_{Seq}$, where $\theta \in [0, 1]$ is the parameter balancing the two losses. We studied the effect of $\theta$ on sequence recovery of the fine-tuned ProteinMPNN model. On one hand, we want $\theta$ to be as high as possible to focus on learning flexibility awareness; on the other hand, we still need the fine-tuned inverse folding model to retain the sequence recovery so that it still has its inverse folding ability. We observed a drop of only one percentage point in terms of sequence recovery, when we trained with $\theta = 0.8$, with the recovery further dropping for higher $\theta$, see Table 8. Therefore, we selected $\theta = 0.8$ as the parameter that still sufficiently maintains the original performance of the vanilla ProteinMPNN.

When balancing the $\mathcal{L}_{Flexpert}$ loss, we also tried to "effectively normalize" the $\mathcal{L}_{Flex}$ and $\mathcal{L}_{Seq}$ losses to have better control with the parameter $\theta$. This we did by normalization of the gradients in the backward pass and then using the parameter $\theta$ to mix at the level of gradients instead of the level of losses. Such training, however, exhibited high instability, so we did not use it for the final version.

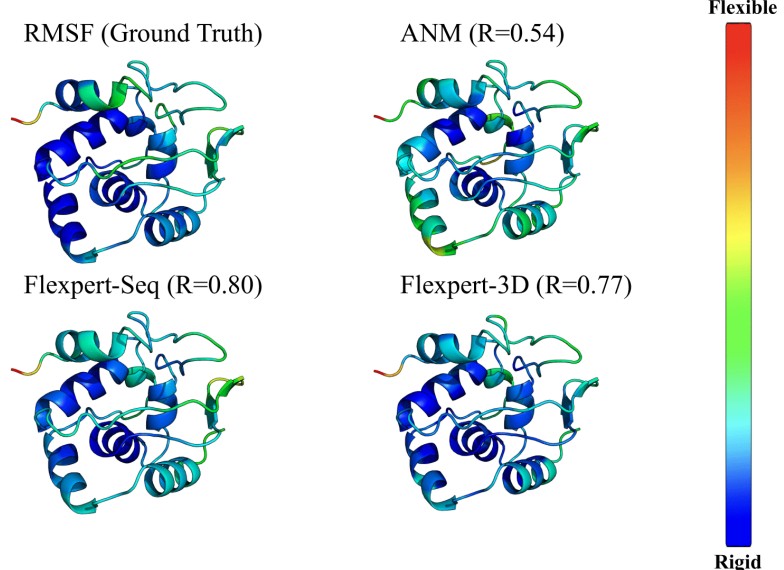

Figure 5: ATLAS datapoint 1c52_A. Anisotropic Network Models perform relatively poorly, which likely propagates to Flexpert-3D causing it to underperform (Pearson R = 0.77) with respect to Flexpert-Seq (Pearson R = 0.8).

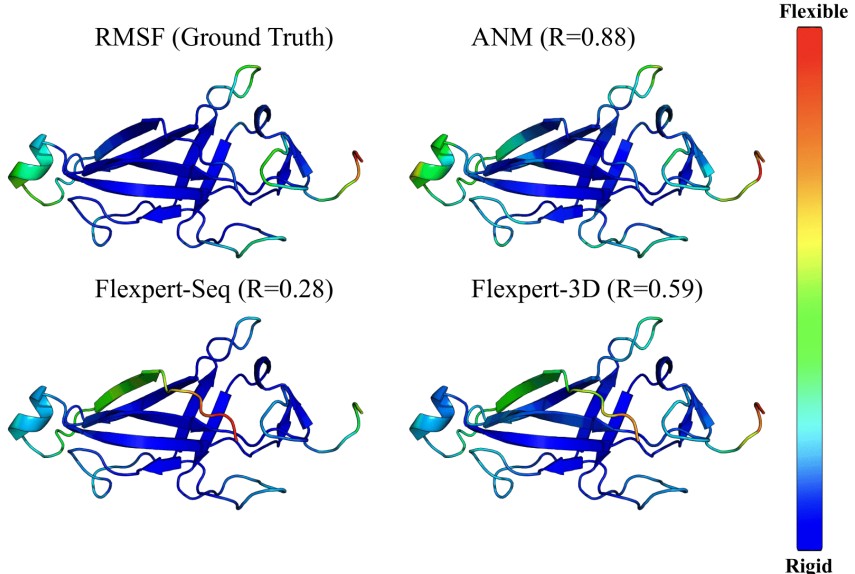

Figure 6: ATLAS datapoint 6o2v_A. This is a failure case for Flexpert-Seq (Pearson R = 0.28), which predicts one beta strand together with the terminal coil to be more flexible than what was observed in MD simulation (see RMSF subfigure). Anisotropic Network Models perform better in this case, possibly helping Flexpert-3D improve over Flexpert-Seq.

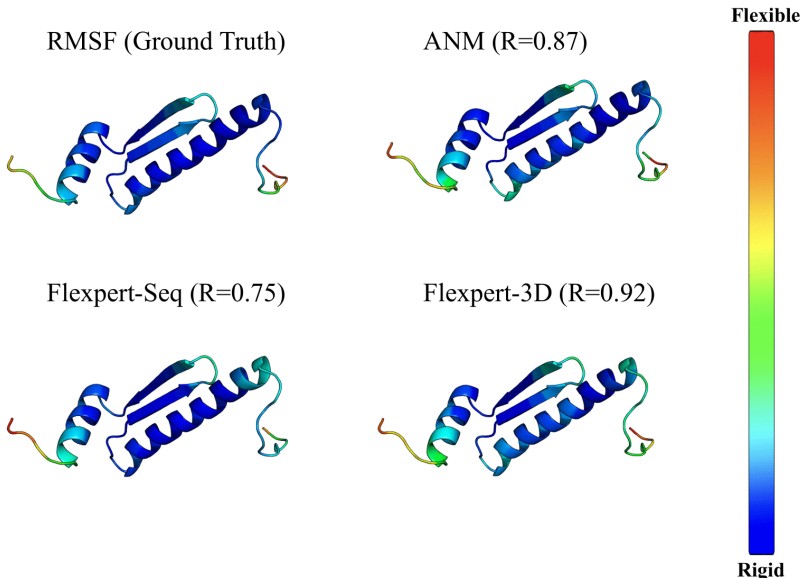

Figure 7: ATLAS datapoint 1egw_B. This datapoint features relatively high variance in the flexibility, posing a challenge for Flexpert-Seq (Pearson R = 0.75). The Anisotropic Network Models work better (Pearson R=0.87) and presumably help Flexpert-3D to excel (Pearson R=0.92).

Table 8: Effect of the parameter $\theta$ for mixing of sequence recovery and flexibility awareness on the sequence recovery of the fine-tuned ProteinMPNN model. The vanilla ProteinMPNN is denoted as $\mathcal{P}$. $\mathcal{P}_F^{\theta=X}$ denotes Flexpert-Design trained with $\theta$ value of $X$.

|  | $\mathcal{P}$ | $\mathcal{P}_F^{\theta=0}$ | $\mathcal{P}_F^{\theta=0.8}$ | $\mathcal{P}_F^{\theta=0.9}$ | $\mathcal{P}_F^{\theta=0.95}$ |
|---|---|---|---|---|---|
| **Sequence recovery** | 0.49 | 0.48 | 0.48 | 0.47 | 0.47 |

**Training and fine-tuning.** We tried to train Flexpert-Design from scratch using the $\mathcal{L}_{Flexpert}$ loss, but it seemed that the $\mathcal{L}_{Flex}$ loss was not informative in the early stages. Therefore, we decided to first train ProteinMPNN in a standard way for 100 epochs on topology split CATH4.3 dataset. Second, we finetuned the model for 12 hours on 1 GPU using the $\mathcal{L}_{Flexpert}$, which resulted in additional 8-11 epochs.

## I    EVALUATION OF STRUCTURE PRESERVATION IN FLEXPERT-DESIGN GENERATED SEQUENCES WITH INCREASED FLEXIBILITY

To evaluate how Flexpert-Design alters the structure of the proteins by the engineering of the increased flexibility, we predicted the structures of the sequences with engineered flexibility using AlphaFold2 inside Colabfold (Jumper et al., 2021; Mirdita et al., 2022). We compared the predicted backbone structures with the input backbone structure and evaluated the confidence of the structure prediction. The experiment was run for 700 randomly selected proteins from the CATH test set, for which we ran the Flexpert-Design predictions with the flexibility increasing instructions and the segment length S=50 (the same choice as in the experiment reported in Figure 3 and Table 4).

The predicted sequences were compared to the ground truth structures at the level of $C_\alpha$ using the root mean square deviation between the structures (RMSD) (Table 9). Furthermore, the pLDDT confidence overall across the whole protein and separately inside and outside the engineered region (where the flexibility increasing instructions were provided) was analyzed, see Table 9.

We see that the Flexpert-Design generated sequences show lower recovery of the ground truth structure compared to ProteinMPNN, which could suggest that Flexpert-Design produces less stable structures or that it introduces notable conformational changes. A particularly significant drop in

Table 9: Evaluation of the Flexpert-Design generated protein sequences using the alignment of the folded backbone structures to the ground truth backbone structures (RMSD column) and the AF2 predicted pLDDT of the whole protein, the engineered region (eng. region) and of the outside of the engineered region (non-eng. region).

| Model | RMSD (Å) ↓ | ø pLDDT ↑ (whole protein) | ø pLDDT ↑ (eng. region) | ø pLDDT ↑ (non-eng. region) |
|---|---|---|---|---|
| ProteinMPNN | 2.00 | 0.77 | 0.81 | 0.73 |
| Flexpert-Design (no loss) | 3.31 | 0.61 | 0.60 | 0.62 |
| Flexpert-Design (ours) | 3.36 | 0.60 | 0.58 | 0.63 |

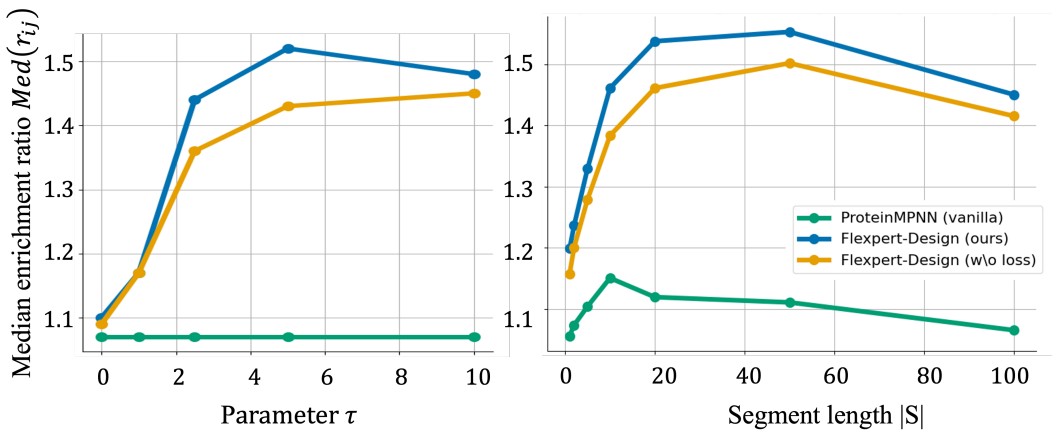

Figure 8: The effect of the flexibility increasing parameter $\tau$ (left) on the median enrichment ratio of our Flexpert-Design model and its variant trained without $\mathcal{L}_{Flex}$ compared to the vanilla Protein-MPNN as a baseline. The effect of the length of the engineered segment on the median enrichment ratio (right).

the pLDDT is in the engineered regions (which tended to be well-structured in the ground truth, as suggested by the high pLDDT for ProteinMPNN sequences), which might hint at the increased flexibility (which was shown by our predictor in Figure 3 and Table 4) causing the drop in pLDDT and increase in RMSD.

## J   THE EFFECT OF FLEXIBILITY INSTRUCTION VALUE AND THE SIZE OF THE ENGINEERED SEGMENT ON FLEXIBILITY ENRICHMENT

The effect of the size of the engineered segment $|S|$ and of the parameter $\tau$ used for increasing the flexibility is presented in Figure 8. The graphs suggest that our model achieves the best median enrichment ratio in the setting of $|S| = 50$ and $\tau = 5$. At the same time, for parameters within the range $\tau \in [2.5, 10]$ and $|S| \in [10, 100]$, the results of our model are comparable to the optimal setting demonstrating reasonable robustness of the model to setting these parameters.

## K   LIMITATION: ENGINEERING DECREASE IN FLEXIBILITY

Analogically to the flexibility-increasing experiment presented in Section 5.2, we also investigated the possibility of decreasing flexibility using our approach with parameter $\tau < 0$. We present the results for $|S| = 50$ and $\tau = -10$ in Table 10.

Table 10: Comparing the ability of flexibility engineering models to decrease the flexibility of engineered segments. The presented median enrichment ratios $Med(r_{ij})$ and the proportion of flexibility decreasing mutations were obtained using the CATH4.3 test set with engineered segments of length $|S| = 50$, and flexibility engineering instructions corresponding to native flexibility incremented by $\tau = -10$ in the engineered segment. The best results in each category are highlighted in bold, second best are underlined.

| | Median enrich. ratio r $\downarrow$ | Prop. of flexibility decreasing mutations $\uparrow$ |
|---|---|---|
| **Flexpert-Design (MSE loss)** | 1.06 | 0.40 |
| **Flexpert-Design (L1 loss)** | **0.99** | **0.51** |
| **Flexpert-Design (w/o loss)** | 1.04 | 0.46 |
| **ProteinMPNN (vanilla)** | 1.07 | 0.39 |

Table 10 also presents the variant of Flexpert-Design trained using L1 loss instead of MSE, which seems to work better for the case of decreasing flexibility. However, despite achieving a modest decrease in flexibility, we consider this task as a limitation of Flexpert-Design. Our explanation is that Flexpert-Design is biased toward positive flexibility engineering instructions because it is trained on the native distribution of flexibilities, which are mostly positive as they approximate the RMSF quantity, which is by definition non-negative. Similarly, we suspect that the Flexpert-3D predictor might be slightly biased toward overpredicting flexibility since it predicts a flexibility increase even for the vanilla ProteinMPNN model. The possible reason is the effect of outliers in the training set. Outliers can exist in the dataset for positive values, but cannot exist for negative due to the non-negative nature of RMSF. Such disbalance can potentially skew the learning distribution toward higher values.

