# OpenReview forum: "Learning to engineer protein flexibility"
_ICLR.cc/2025/Conference — ICLR 2025 Poster_

### Official Review · Reviewer_R4S7 · 2024-10-23

**Soundness:** 2
**Presentation:** 3
**Contribution:** 2
**Rating:** 6
**Confidence:** 3

**Summary:**

The manuscript presents an deep learning based approach to predict protein flexibility and engineer protein sequence towards more flexibility. There are three steps in the manuscript. The first step is to compare metrics that quantify protein flexibility. The second step is to repurpose pre-trained protein language model for flexibility prediction. Here, the authors trained Flexpert-Seq (no 3D structure input) and Flexpert-3D (with 3D structure input) for the regression task. Last but not the least, the authors proposed a framework to train ProteinMPNN with flexibility instructions to generate more flexible protein sequence given the a input 3D structure.

**Strengths:**

The idea of predicting the flexibility of protein given structure or sequence is rather novel. The comparison between the metrics for evaluating protein flexibility is valuable. The framework proposed to inverse fold for better flexibility, Flexpert-Design, can be valuable for specific applications requiring more flexible proteins.

**Weaknesses:**

1. Writing can be improved. For example, in line 79-80, "From these simulations, Root Mean Square Fluctuations (RMSF)." is not a full sentence. Presentation can also be improved. In table 3 and 4, only MD results are bolded, whereas the ANM (MD str.) results are not. Authors can also add arrows to indicate if each metric is the higher the better or the lower the better.
2. The benchmark needs more explanation, especially the flexibility prediction one. It is not clear to me how AF2 and ESMFold are used for flexibility prediction (the regression task). The interpretation of the benchmarking results (table 3 and 4) is also vague.
3. There is not enough elaboration on why more flexible protein is needed. One major contribution of this paper is the Flexpert-Design framework, which tunes ProteinMPNN to generate more flexible protein given the prediction of a flexibility predictor. However, it is not clear to me (from reading the manuscript) the purpose of modifying the flexibility of a protein. The authors can focus more on the motivation.
4. The Flexpert-Design can introduce bias from Flexpert-3D into the ProteinMPNN model through the flex loss. Also, the correlation between the input structure and the flexibility instruction can be low when the flexibility instruction is artificially increased, i.e. the input structure can never have high flexibility given by the flexibility instruction. In such case, the prediction of both flexibility instructed ProteinMPNN and Flexpert-3D can be unreliable.

**Questions:**

1. How is RMSF calculated? It would be better to note the RMSF equation in the manuscript.
2. Line 152-153, author claims higher pLDDT uncertainty indicates lower flexibility, which is kind of counter-intuitive. Can authors elaborate?
3. When benchmarking the flexibility prediction ability, how are AF2 and ESMFold used? Based on my understanding of the manuscript, the flexibility prediction is a regression task. Did authors attach a prediction head to baseline models? Please elaborate.

---

> ### Author Response · Authors · 2024-11-22
> **Response to Reviewer R4S7 (Part 1/5)**
>
> We thank the reviewer for their valuable feedback, which significantly helps improve the clarity of our manuscript. Below, we address the individual points raised among ‘Weaknesses’ and ‘Questions’. We changed the numbering from the original review to “W1”-”W4” and “Q1”-”Q3” to label the weaknesses and the questions, respectively. We then use these labels for cross-references in case of related questions and weaknesses.
>
> > **W1:** Writing can be improved. For example, in line 79-80, "From these simulations, Root Mean Square Fluctuations (RMSF)." is not a full sentence. Presentation can also be improved. In table 3 and 4, only MD results are bolded, whereas the ANM (MD str.) results are not. Authors can also add arrows to indicate if each metric is the higher the better or the lower the better.
>
> We thank the reviewer for pointing out these issues.
>
> We have corrected the incomplete sentence to:
>
> *“From these simulations, Root Mean Square Fluctuations (RMSF) can be computed per residue to quantify their flexibility as observed in the simulations.”*
>
> In Tables 3 and 4, only the results in the ‘MD’ row are highlighted because the correlation to the MD simulations was considered as the measure of success, whereas the correlation to the ANM is reported as a reference, to show how independent of ANM the predictions of Flexpert-Seq and Flexpert-3D are. We apologize for the confusing caption of Table 3, which we now clarify by changing the sentence
>
> *“Also, our approach outperforms the AlphaFold2 and ESMFold baselines and obtains the best correlation with the flexibility estimates obtained from Molecular Dynamics (row MD) and Anisotropic Network Models (row ANM).”*
>
> to
>
> *“Our approach outperforms the AlphaFold2 and ESMFold baselines and obtains the best correlation with the flexibility estimates obtained from Molecular Dynamics (see the value in bold). Interestingly, the correlation coefficient of our approach and Anisotropic Network Models (ANM) is just 0.66, suggesting that both our approach and ANM might be complementary, motivating the development of a predictor combining the sequence-based predictor with an ANM.”*
>
> At the end of the caption of Table 4, we have added the following sentence:
>
> “*Note that the correlation coefficient of our model with ANM is 0.81 (see the bottom row), indicating that our model relies on ANM input only partially and actually learns novel information from the sequence.*"
>
> We appreciate the tip on including arrows to indicate whether maximization or minimization is desired. After careful consideration, we have decided to specify this information directly in the caption—while we want to maximize the correlation to MD, it is not so clear for the correlation to ANM, since we want our methods to bring more than what is covered by ANM.
>
> > **W2:** The benchmark needs more explanation, especially the flexibility prediction one. It is not clear to me how AF2 and ESMFold are used for flexibility prediction (the regression task). The interpretation of the benchmarking results (table 3 and 4\) is also vague.
>
> We use the confidence modules of AF2 and ESMFold to predict pLDDT, which we use as a proxy for flexibility scores, as detailed in Section 3.1 “Methods for quantification of protein flexibility”, lines 150-156 (AF2) and lines 157-159 (ESMFold) respectively.
>
> We thank the reviewer for identifying a possible source of confusion. To clarify the details of the flexibility metrics reported in Table 1, we have added a specific pointer to Section 3.1 into the caption of the table:
>
> *Table 1: Pearson correlation coefficients of flexibility predictions obtained by PDB B-factors and by computational methods (**see Section 3.1 for details of the studied flexibility quantification methods**). The reported coefficients are averaged over the 1390 proteins from the ATLAS dataset. As the table is symmetric, we show values only in the upper triangle. The best result in terms of correlation MD simulations is highlighted in bold, the second best is underlined.*
>
> To improve the interpretation of the benchmarking results, we have expanded the captions of Tables 3 and 4, as described in the response to 'W1'.

---

> ### Author Response · Authors · 2024-11-22
> **Response to Reviewer R4S7 (Part 2/5)**
>
> > **W3:** There is not enough elaboration on why more flexible protein is needed. One major contribution of this paper is the Flexpert-Design framework, which tunes ProteinMPNN to generate more flexible protein given the prediction of a flexibility predictor. However, it is not clear to me (from reading the manuscript) the purpose of modifying the flexibility of a protein. The authors can focus more on the motivation.
>
> We thank the reviewer for pointing out that the motivation for the engineering of protein flexibility should be described better. To address this, we have expanded the statement in the Introduction:
>
> *“Proteins are highly dynamic biomolecules and the presence of flexible regions is critical for their biological function (Corbella et al., 2023; Lemay-St-Denis et al., 2022).”*
>
> with more concrete examples illustrating the importance of flexibility:
>
> *“In particular, fine-tuning conformational dynamics of loops close to the active site has been shown to be an important method for modulating substrate specificities ([Romero-Rivera et al. 2022](https://doi.org/10.1021/jacsau.2c00063)), turnover rates ([Crean et al. 2021](https://doi.org/10.1021/jacs.0c11806)), and pH dependency ([Shen et al. 2021](https://doi.org/10.1021/jacsau.1c00054)) of enzymes. In addition, the biological function of many proteins often requires that a small molecule is transported through their structures, for example via tunnels leading to the active site, whose dynamical properties are thus critical for protein function ([Jurcik et al. 2018](https://pubmed.ncbi.nlm.nih.gov/29741570/)). Modulating the flexibility of protein parts even far away from the active sites has been shown to be an efficient strategy for improving proteins ([Karamitros et al. 2022](https://doi.org/10.1073/pnas.2118979119)).”*
>
> In response to ‘Q2’ raised by  Reviewer i3SF, we have also provided the following extra motivation for engineering of flexibility:
>
> The direct improvement of protein function by engineering its flexibility is not universal, but there are proteins where flexibility is believed to be directly linked to their function. It has been previously shown that the dynamic behavior of a bifunctional haloalkane dehalogenase/luciferase system affects the main activity output ([Schenkmayerova et al. 2021](https://www.nature.com/articles/s41929-022-00895-z), [Schenkmayerova et al. 2023](https://www.nature.com/articles/s41467-021-23450-z)). In protein engineering this has previously been addressed by a protocol to guide flexibility design by loop transplant ([Planas-Iglesias et al. 2022](https://academic.oup.com/nar/article/50/W1/W465/6570728)). Loop grafting in association to engineering protein dynamics has also been explored in beta-lactamases ([Gobeil et al. 2014](https://pubmed.ncbi.nlm.nih.gov/25200606/), [Park et al. 2006](https://www.science.org/doi/10.1126/science.1118953)), isomerases ([Romero-Rivera 2022](https://pubs.acs.org/doi/10.1021/jacsau.2c00063)), and tyrosyne phosphatases ([Crean et al. 2024](https://www.cambridge.org/core/journals/qrb-discovery/article/sequence-dynamics-function-relationships-in-protein-tyrosine-phosphatases/9A8C55E63F9852869D3A21B742705B79), [Shen et al. 2022](https://pubs.rsc.org/en/content/articlelanding/2022/sc/d2sc04135a)). In all these cases the enzymatic activity was modulated by flexibility in regions far from the catalytic site. Thus the engineering of their flexibility can lead to functional improvements.

---

> ### Author Response · Authors · 2024-11-22
> **Response to Reviewer R4S7 (Part 3/5)**
>
> > **W4:** The Flexpert-Design can introduce bias from Flexpert-3D into the ProteinMPNN model through the flex loss. Also, the correlation between the input structure and the flexibility instruction can be low when the flexibility instruction is artificially increased, i.e. the input structure can never have high flexibility given by the flexibility instruction. In such case, the prediction of both flexibility instructed ProteinMPNN and Flexpert-3D can be unreliable.
>
> We thank the reviewer for raising this point. We have run an experiment to see the effect of Flexpert-Design on the structure of the proteins and confidence of the structure prediction:
>
> We ran an experiment with 700 randomly selected proteins from the CATH testset, for which we ran the Flexpert-Design predictions with the flexibility increasing instructions $\tau=5$ and the segment length S=50 (the same choice as in the experiment reported in Figure 4). For the Flexpert-Design predicted sequences, we ran the AlphaFold2 structure prediction (using the default settings of [localcolabfold](https://github.com/YoshitakaMo/localcolabfold)) and we compared the predicted structures of the predicted sequences with the ground truth structures at the level of $C_\alpha$ to compute the root mean square deviation between the structures (RMSD). Furthermore, we looked at the AF2 pLDDT confidence overall across the whole protein and separately inside and outside the engineered region (where we provided the flexibility increasing instructions). We present the results of the experiment in the table below (Table 9\) and we will include it in the Appendix of the manuscript:
>
> Table 9: Evaluation of the Flexpert-Design generated protein sequences using the AlphaFold2 pLDDT and the alignment of the folded backbone structures to the ground truth backbone structures (at the $C_\alpha$ level).
>
> | Model | RMSD  to Ground Truth ↓ (Å)  | ø pLDDT ↑ (whole protein) | ø pLDDT ↑ (engineered region) | ø pLDDT ↑ (outside engineered region) |
> | :---- | :---- | :---- | :---- | :---- |
> | ProteinMPNN | 2.00 | 0.77 | 0.81 | 0.73 |
> | Flexpert-Design (no loss) | 3.31 | 0.61 | 0.60 | 0.62 |
> | Flexpert-Design (ours) | 3.36 | 0.60 | 0.58 | 0.63 |
>
> We see that the Flexpert-Design generated sequences show lower recovery of the ground truth structure compared to ProteinMPNN, which could suggest that Flexpert-Design produces less stable structures or that it introduces notable conformational changes. A particularly significant drop in the pLDDT is in the engineered regions (which tended to be well-structured in the ground truth, as suggested by the high pLDDT for ProteinMPNN sequences), which might hint at the increased flexibility (which was shown by our predictor in Figure 4 and Table 5) causing the drop in pLDDT and increase in RMSD.

---

> ### Author Response · Authors · 2024-11-22
> **Response to Reviewer R4S7 (Part 4/5)**
>
> > **Questions:**
> > **Q1:** How is RMSF calculated? It would be better to note the RMSF equation in the manuscript.
>
> We thank the reviewer for bringing up this point, as RMSF is a key quantity in our work. We will elaborate on the computation of RMSF in the Appendix and include a pointer to this new Appendix F in the main text where we introduce RMSF (after the previously incomplete statement at lines 79-80 of the submission). In the new Appendix F, we will present the following description how the RMSF values in our manuscript were obtained, including the formulas for both the cases when MD simulation or Elastic Network Models (ENM) were used for the estimation of the flexibility:
>
> *“The Root Mean Square Fluctuations (RMSF) are a way of quantifying protein flexibility, typically based on simulated data of dynamics or based on simplified models of dynamics such as the Elastic Network Models (ENM). In this work, we use RMSF computed on top of both MD simulations and ENMs (Anisotropic Network Models and Gaussian Network Models). These RMSFs were obtained in the following way:*
>
> ***RMSF from MD simulations.** We used the values provided directly with the ATLAS dataset, which provides the per residue RMSF for each of its simulation replicas. To represent each protein's flexibility profile, we used the per residue RMSF averaged over all 3 simulation replicas of each protein. The values provided by the ATLAS dataset were computed as standard deviation of the α-carbon displacements, using the GROMACS software. That is, the RMSF for MD simulations were computed using the following formula:*
>
>
>
>
> $$
> \rho_i = \sqrt{\langle || r_i- \langle r_i \rangle ||^2_2 \rangle}   ,
> $$
>
> *where i  denotes the RMSF of residue i, $r_i$ stands for the coordinates of the α-carbon of residue i in the global reference frame, the angle brackets  〈  〉 stand for the average over all the frames of the simulation and $|| \cdot ||_2$ denotes the Euclidean norm.*
>
> ***RMSF for Elastic Network Models (Anisotropic Network Models).** We used the ProDy ([Bakan et al. 2011](https://pubmed.ncbi.nlm.nih.gov/21471012/)\) package to compute the ENMs and the corresponding RMSF values. Firstly, we used ProDy to build the Hessian matrix, based on the α-carbon coordinates of our input structure and the cutoff of 16 Angstrom, then the non-trivial modes of the Hessian were calculated, and consequently these modes were used to solve the problem of finding the inverse of the Hessian to compute the fluctuations according to the formula for the ENMs:*
>
> $$
> \rho_i = \sqrt{\frac{1}{C}\mathrm{Tr}([H^{-1}]_{ii})}  ,
> $$
>
> *where i  denotes the RMSF of residue i, C denotes a constant that is typically equal to the product of the Boltzmann constant and the temperature ($C=k_{B}T$), but for us this is arbitrary as we are interested in the correlations of the fluctuations and the correlations are not affected by the scaling by a constant. The $\mathrm{Tr}$ in the formula stands for the trace of the matrix $[H^{-1}]_{ii}$ which is the 3x3 submatrix of the inverse Hessian corresponding to the residue i (the 3x3 submatrix comes from the fact that the $\alpha$-carbon positions are represented in 3D).*
>
> ***RMSF for Elastic Network Models (Gaussian Network Models).** For Gaussian Network models, the procedure is analogous to the case of Anisotropic Network Models, with only the difference that instead of Hessian matrix, the Kirchoff matrix is used:*
>
> $$
> \rho_i = \sqrt{\frac{1}{C}\mathrm{Tr}([\Gamma^{-1}]_{ii})}  ,
> $$
>
> *where $\Gamma$ is the Kirchhoff matrix constructed for the α-carbon coordinates of the input structure considering the cutoff of 16 Angstrom.*
>
> *More details on calculation of fluctuations from ENMs (both Anisotropic and Gaussian Network Models) can be found in ([Yang et al. 2009](https://pmc.ncbi.nlm.nih.gov/articles/PMC2718344/)).”*
>
> > **Q2:** Line 152-153, author claims higher pLDDT uncertainty indicates lower flexibility, which is kind of counter-intuitive. Can authors elaborate?
>
> We thank the reviewer for identifying this confusing nomenclature.  We have modified the manuscript to refer to the pLDDT score as “pLDDT confidence” (rather than "pLDDT uncertainty"). The pLDDT score, as defined for example in AF2, is in the interval \[0,1\], where the 0 and 1 values correspond to, respectively, absolutely uncertain and fully confident predictions of the structure. While a low confidence might be due to the error of the model (for example because the model has not seen relevant examples in its training set), it can also be caused by uncertainty in the ground truth structure if that structure is flexible. A higher pLDDT score (i.e., a higher pLDDT confidence) can thus indicate a more rigid structure and lower flexibility.

---

> > ### Comment · Reviewer_R4S7 · 2024-11-25
> >
> > Thanks, switching from "pLDDT uncertainty" to "pLDDT confidence" makes sense.

---

> > > ### Comment · Reviewer_R4S7 · 2024-11-25
> > >
> > > Thanks for the thorough response to my questions. I believe the presentation of paper has been improved after revision. I'll increase the rating.

---

> > > > ### Author Response · Authors · 2024-11-26
> > > >
> > > > We thank the reviewer for the acknowlegment of the improvements in our manuscript achieved during revision and based on the valuable feedback provided by the reviewers.

---

> ### Author Response · Authors · 2024-11-22
> **Response to Reviewer R4S7 (Part 5/5)**
>
> > **Q3:** When benchmarking the flexibility prediction ability, how are AF2 and ESMFold used? Based on my understanding of the manuscript, the flexibility prediction is a regression task. Did authors attach a prediction head to baseline models? Please elaborate.
>
> We thank the reviewer for the question. We used the confidence modules of AF2 and ESMFold to predict the pLDDT, which we used as a proxy for the flexibility prediction. We discuss this further in our response to ‘W2’ (see above).

---

### Official Review · Reviewer_DNnV · 2024-11-03

**Soundness:** 2
**Presentation:** 3
**Contribution:** 3
**Rating:** 8
**Confidence:** 4

**Summary:**

In this paper the authors investigate the correlation of various in silico proxies to root mean square fluctuations of protein structures derived from molecular dynamics calculations. They posit that augmenting inverse folding methods for protein design to guide for RMSF as a proxy for flexibility is desirable for design applications. To this end, they introduce Flexpert-Seq, Flexpert-3D, and Flexpert-Design. They demonstrate that supervised models outperform other methods for predicting RMSF from sequence alone, and that they can guide inverse folding design to favor more flexible residues.

**Strengths:**

The paper introduces an interesting problem, explores the problem and relevant dataset before diving into the methodology, and shows that simulated dynamical properties of proteins can be predicted from sequence alone, or sequence and simple structural featurizations, offering important advantages in speed and use-of-use compared to other methods.
The discussion of sequence recovery as an incomplete evaluation of inverse folding methods, and the predominant usage of ProteinMPNN due to its extensive wet lab validation, is appreciated.
The Table 3 and 4 results are nice, clear demonstrations that the authors’ hypothesis is correct, and that a supervised model outperforms more indirect methods like pLDDT and ANM for predicting RMSF values.

**Weaknesses:**

The authors might provide a concise definition of “flexibility” earlier on in the paper. Flexibility could mean many things and be quantified and measured different ways, experimentally and in silico, as detailed in Section 3.1.
Some discussion is needed of the incompleteness of the methods introduced in Section 3.1. pLDDT and B-factors can be confounded by many other covariates that are not flexibility, and RMSF calculations from MD may not accurately reflect the real world biophysics of dynamic proteins in a physiological context, where interactions with solvent and other biomolecules will affect effective flexibility.
A small nit, say “MD trajectories” rather than MDs to clarify the source of reference datasets.
I would recommend capturing all the newly introduced methods under subheadings under a Methods section heading, and then going into results. With the current structure I expected to see Results after Section 4, but Section 5 introduces another method.
Without simple baselines (some ideas provided in the Questions below) it is difficult to assess the results presented in the design section.
The Conclusions section restates what was done, which is covered in the intro. Discussion of limitations and future directions would be appreciated.

**Questions:**

Can the authors support the statement in the abstract that accounting for flexibility is the major limitation of current protein design methods? There are many limitations of current methods; flexibility is certainly an important one, but it is not clear that it is a singular limitation.
Can the authors show some failure cases where predictions from Flexpert-Seq do not agree well on the ATLAS dataset? I’m curious if there are certain classes of protein or biophysical features that are difficult to predict well.
The enrichment for A, S, T, and G residues and depletion of bulkier residues on flexibility guidance makes sense - is there a “self-consistency” check that ensures that the guided designs retain the desired predicted backbone structure? How does Flexpert-design compare to running ProteinMPNN and using protein language model likelihoods to identify residues where mutations to smaller, more flexible residues are likely to be tolerated?
Relatedly, per-residue RMSF seems to be a “local” notion of flexibility, which causes a straightforward enrichment of a few specific, small residues. Can the authors imagine ways to include a more non-local measure of flexibility that may induce larger or smaller fluctuations across a biomolecule, without dramatically changing the conditional amino acid distribution?

---

> ### Author Response · Authors · 2024-11-22
> **Response to Reviewer DNnV (Part 1/3)**
>
> We thank the reviewer for their very detailed and constructive feedback, which helps improve the clarity and overall quality of our manuscript. Below, we address the reviewer's suggestions. To this end, we split the ‘Weaknesses’ and the ‘Questions’ into individual points and label them as “W1”-”W5” and “Q1”-”Q5”, respectively. We then provide our response to each of the points.
>
>
> > **W1:** The authors might provide a concise definition of “flexibility” earlier on in the paper. Flexibility could mean many things and be quantified and measured different ways, experimentally and in silico, as detailed in Section 3.1.
>
> We thank the reviewer for the suggestion. To provide a concise definition of flexibility, we modified the beginning of Section 3.1 in the following way (new text is in bold):
>
> *“**In this work we focus on flexibility as a quantity describing the ability of individual residues (mainly their backbone) to move within the protein structure. To this end, we introduce in this section the** relevant methods for quantification of protein flexibility, and in particular, the metrics used with each method for reporting per-residue protein flexibility values.”*
>
> We discuss the caveats of the individual techniques for quantification and measurement of flexibility in more detail in the response to ‘W2’ below.
>
> > **W2:** Some discussion is needed of the incompleteness of the methods introduced in Section 3.1. pLDDT and B-factors can be confounded by many other covariates that are not flexibility, and RMSF calculations from MD may not accurately reflect the real world biophysics of dynamic proteins in a physiological context, where interactions with solvent and other biomolecules will affect effective flexibility. A small nit, say “MD trajectories” rather than MDs to clarify the source of reference datasets.
>
> We thank the reviewer for this insightful comment. We present the following discussion which we will incorporate into the manuscript:
>
> *“While it is true that pLDDT is a direct measure of estimated uncertainty by AlphaFold (that may be induced by sequence variability in the region, for instance), it has been shown to be negatively correlated with flexibility ([Saldano et al. 2022](https://pubmed.ncbi.nlm.nih.gov/35561203/)) and exploited to approximate flexibility as measured by Nuclear Magnetic Resonance (NMR) ([Ma et al. 2023](https://onlinelibrary.wiley.com/doi/full/10.1002/prot.26471)). B-factors from crystallographic data come with their own limitations. The crystallographic B-factor describes the inherent thermal vibrations of atoms in a crystal lattice. In such an organized environment, the natural movement of the protein is diminished, and artifacts (i.e. crystal packing) may lead to an underestimation of the real protein motions ([Eyal et al. 2005](https://www.sciencedirect.com/science/article/pii/S0022283605006339)). MD trajectories are also limited in considering an isolated system (of one or more proteins) in a given modeled solvent environment. The first limitation comes from the fact that the effect of the solvent on the molecule is yet approximate. However, these interactions typically affect the vibrational states of amino-acid side chains that are not considered by the RMSFs obtained from the ATLAS dataset, since these RMSFs were only calculated from $C_\alpha$ atoms. Second, potential interactions driven by other molecules not present in the system that might affect the protein flexibility are thus neglected. These interactions are countless and need to be addressed case-by-case in individual MD simulations. While capturing the direct or allosteric effects of other interactors (be polymers or small molecules) is beyond our study, the ATLAS dataset and mdCATH are valuable resources to access comprehensive MD simulation data representing a large variety of protein folds. Understanding and capturing protein flexibility in those general cases is in the scope of the present manuscript.**“***
>
> We will furthermore reflect in the manuscript the reviewer's point on the nomenclature and we will properly use the term MD trajectories instead of our previously used informal term.
>
> > **W3:** I would recommend capturing all the newly introduced methods under subheadings under a Methods section heading, and then going into results. With the current structure I expected to see Results after Section 4, but Section 5 introduces another method.
>
> We thank the reviewer for the recommendation. In the revised manuscript, we will merge the current Sections 4 and 5 to have all newly introduced methods in the same Methods section.

---

> ### Author Response · Authors · 2024-11-22
> **Response to Reviewer DNnV (Part 2/3)**
>
> > **W4:** Without simple baselines (some ideas provided in the Questions below) it is difficult to assess the results presented in the design section.
>
> We thank the reviewer for raising this point. We address this point in response to ‘Q4’ below.
>
> > **W5:** The Conclusions section restates what was done, which is covered in the intro. Discussion of limitations and future directions would be appreciated.
>
> We thank the reviewer for this suggestion. We will add a summary of the limitations (so far only discussed in the Appendix) directly to the conclusions and we will discuss the future directions there as well. If there is not enough space eventually for the whole section, we will shorten the summary of the paper's contributions. We propose to modify the end of the Conclusions by the following discussion (new text is in bold):
>
> "*Thirdly, we propose a learning strategy Flexpert-Design, which uses the Flexpert-3D flexibility predictor to steer an inverse folding model toward a generation of sequences with increased flexibility. We demonstrate that inverse folding models can be steered toward generating protein sequences with increased flexibility as measured by our protein flexibility predictor. **In some practical protein engineering tasks, it would also be of interest to engineer protein sequences with decreased flexibility, where the ability of our Flexpert-Design model seems limited (see Appendix E). This motivates future development of new models, which would provide a tighter control of the designed sequence’s flexibility and which would be successful regardless of whether the flexibility was instructed to increase or decrease. An interesting future direction could be to use Flexpert-3D for guidance of discrete flow matching models.***"
>
> > **Questions:**
> > **Q1:** Can the authors support the statement in the abstract that accounting for flexibility is the major limitation of current protein design methods? There are many limitations of current methods; flexibility is certainly an important one, but it is not clear that it is a singular limitation.
>
> We thank the reviewer for pointing this out. We have moderated and clarified this claim by changing the original sentence:
>
> *“However, their major limitation is the inability to account for protein flexibility, a property crucial for protein function.”*
>
> To
>
> *“However, the current methods mostly focus on the design of proteins with a fixed backbone structure, which leads to their limited ability to account for protein flexibility, one of the crucial properties for protein function.”*
>
> > **Q2:** Can the authors show some failure cases where predictions from Flexpert-Seq do not agree well on the ATLAS dataset? I’m curious if there are certain classes of protein or biophysical features that are difficult to predict well.
>
> We thank the reviewer for the suggestion. We will try to identify the relevant failure modes and add the visualization of interesting failure cases to the edited Appendix which we will upload later during the rebuttal period, if such an arrangement would be fine by the reviewer.

---

> ### Author Response · Authors · 2024-11-22
> **Response to Reviewer DNnV (Part 3/3)**
>
> > **Q3:** The enrichment for A, S, T, and G residues and depletion of bulkier residues on flexibility guidance makes sense \- is there a “self-consistency” check that ensures that the guided designs retain the desired predicted backbone structure?
>
> We thank the reviewer very much for raising this point. To address the reviewer's suggestion to run a self-consistency check whether the backbone structure is retained, we ran an experiment with 700 randomly selected proteins from the CATH testset, for which we ran the Flexpert-Design predictions with the flexibility increasing instructions $\tau=5$ and the segment length S=50 (the same choice as in the experiment reported in Figure 4). For the Flexpert-Design predicted sequences, we ran the AlphaFold2 structure prediction (using the default settings of [localcolabfold](https://github.com/YoshitakaMo/localcolabfold)) and we compared the predicted structures of the predicted sequences with the ground truth structures at the level of $C_\alpha$ to compute the root mean square deviation between the structures (RMSD). Furthermore, we looked at the AF2 pLDDT confidence overall across the whole protein and separately inside and outside the engineered region (where we provided the flexibility increasing instructions). We present the results of the experiment in the table below (Table 9\) and we will include it in the Appendix of the manuscript:
>
> Table 9: Evaluation of the Flexpert-Design generated protein sequences using the AlphaFold2 pLDDT and     the alignment of the folded backbone structures to the ground truth backbone structures (at the $C_\alpha$  level).
>
> | Model | RMSD  to Ground Truth ↓ (Å)  | ø pLDDT ↑ (whole protein) | ø pLDDT ↑ (engineered region) | ø pLDDT ↑ (outside engineered region) |
> | :---- | :---- | :---- | :---- | :---- |
> | ProteinMPNN | 2.00 | 0.77 | 0.81 | 0.73 |
> | Flexpert-Design (no loss) | 3.31 | 0.61 | 0.60 | 0.62 |
> | Flexpert-Design (ours) | 3.36 | 0.60 | 0.58 | 0.63 |
>
> We see that the Flexpert-Design generated sequences show lower recovery of the ground truth structure compared to ProteinMPNN, which could suggest that Flexpert-Design produces less stable structures or that it introduces notable conformational changes. A particularly significant drop in the pLDDT is in the engineered regions (which tended to be well-structured in the ground truth, as suggested by the high pLDDT for ProteinMPNN sequences), which might hint at the increased flexibility (which was shown by our predictor in Figure 4 and Table 5) causing the drop in pLDDT and increase in RMSD.
>
> > **Q4:** How does Flexpert-design compare to running ProteinMPNN and using protein language model likelihoods to identify residues where mutations to smaller, more flexible residues are likely to be tolerated?
>
> We thank the reviewer for the suggestion of an interesting baseline experiment. We are working on this and we will post the results later during the rebuttal period. We plan to use the likelihoods for the different amino acids extracted directly from ProteinMPNN, unless the reviewer has a specific interest in involving a protein language model in the experiment.
>
> > **Q5:** Relatedly, per-residue RMSF seems to be a “local” notion of flexibility, which causes a straightforward enrichment of a few specific, small residues. Can the authors imagine ways to include a more non-local measure of flexibility that may induce larger or smaller fluctuations across a biomolecule, without dramatically changing the conditional amino acid distribution?
>
> We thank the reviewer for the interesting suggestion. One way to tackle this might be to first take the secondary structure assignments of the residues. Ideally, by training on labels smoothed across the secondary structure elements (SSEs), the model could be trained to focus on the interfaces of neighboring SSEs and to design the flexibility of the connecting hinges. Such a model might tend to preserve the secondary structure elements and their amino acid distribution, tuning the flexibility mostly by mutating the hinges. A more sensitive model than the current version of Flexpert-Design might be needed to achieve this, for example relying on the discrete flow matching models, as suggested in our response to ‘W5’.

---

> > ### Comment · Reviewer_DNnV · 2024-11-24
> >
> > I thank the authors for their careful and comprehensive response. By acknowledging the limitations and scope of the problem, and strengthening their claims with further experiments and analysis, I believe the paper is improved and addresses an under-studied aspect of protein design. I am happy to increase my score.

---

> > > ### Author Response · Authors · 2024-11-24
> > >
> > > We thank the reviewer for acknowledging the improvements in our submission, which were achieved through additional experiments and analysis based on all the constructive reviews.

---

> ### Author Response · Authors · 2024-11-29
> **Qualitative examination of Flexpert-Seq and Flexpert-3D and a new baseline for Flexpert-Design.**
>
> We would like to follow up on our previous response to the review. To answer the questions ‘Q2’ and ‘Q4’, we have performed two additional experiments, which we describe below:
>
> >Q2: Can the authors show some failure cases where predictions from Flexpert-Seq do not agree well on the ATLAS dataset? I’m curious if there are certain classes of protein or biophysical features that are difficult to predict well.
>
> We have examined the predictions of Flexpert-Seq and Flexpert-3D obtained for the ATLAS test set. In Appendix G in the updated manuscript we show four qualitative examples, including a failure case for Flexpert-Seq (Figure 6) and, for contrast, a case where Flexpert-Seq outperformed Flexpert-3D (Figure 5). In general, we found that the correlation of Flexpert-Seq predictions to the ground truth RMSF tends to decrease for longer protein sequences, although the effect is not profound.
>
>
> >Q4: How does Flexpert-design compare to running ProteinMPNN and using protein language model likelihoods to identify residues where mutations to smaller, more flexible residues are likely to be tolerated?
>
> Thank you very much again for this useful suggestion. It is very interesting to compare our method to a baseline utilizing the domain knowledge that smaller residues allow for higher flexibility.
>
> We ran the following experiment:
>
> 1. In the same setting as the one used to report the results of Flexpert-Design in Figure 3 and Table 4, we ran the predictions of ProteinMPNN to obtain the log probabilities of individual amino acids at each position in the sequence.
>
> 2. Outside the engineered region, we sampled the most probable amino acids.
>
> 3. Inside the engineered region, we narrowed down the selection of amino acids to only 5 amino acids: A, S, T, G, and the most probable amino acid other than A, S, T, and G. We then sampled proportionally to the probabilities of these amino acids.
>
> Interestingly, this simple baseline was able to reach the median enrichment ratio of 1.43, matching the performance of the variant of Flexpert-Design trained without the $L_{Flex}$ loss (median enrichment ratio 1.43). However, it still lagged behind the performance of full Flexpert-Design (median enrichment ratio 1.52).

---

### Official Review · Reviewer_i3SF · 2024-11-04

**Soundness:** 3
**Presentation:** 3
**Contribution:** 2
**Rating:** 5
**Confidence:** 2

**Summary:**

the paper address an interesting problem where one wan to generate novel proteins while accounting for protein flexibility—a feature that is critical for protein function but has proven challenging to model and integrate due to data scarcity, heterogeneity, and the high costs of data generation.  The paper provide benchmark dataset and ways to quantify flexibility. They  adapt the inverse folding model for flexibility, guiding protein sequences towards targeted flexibility in specific regions.

**Strengths:**

1. The study provides a robust review of various protein flexibility quantification methods and assesses the utility of diverse data sources.
2. The development of Flexpert-Design to adjust flexibility in specified regions is significant, especially for applications in enzyme engineering, drug design, and biocatalysis.

**Weaknesses:**

1. A major limitation is the lack of extensive experimental validation.

**Questions:**

1. Given the relatively small ATLAS dataset, how does the predictor generalize to other protein datasets? Has the model been tested on proteins outside the ATLAS dataset to evaluate robustness?
2. I am wondering if the proteins generated  by this model (due to the flexibility-aware nature of the model) show evidence of functional improvements (e.g., improved stability, interaction specificity), or is the flexibility change primarily theoretical?
3. The Flexpert loss function combines flexibility and sequence objectives with a mixing parameter, θ. How sensitive is the final model performance to different values of θ, and how was the optimal value chosen?

---

> ### Author Response · Authors · 2024-11-22
> **Response to Reviewer i3SF (Part 1/2)**
>
> We thank the reviewer for their valuable feedback. Below, we address the individual points raised among ‘Weaknesses’ and ‘Questions’. We changed the numbering from the original review to “W1” and “Q1”-”Q3” to label the weakness and the questions, respectively. We then use these labels for cross-references in case of related questions and weaknesses.
>
> > **W1:** A major limitation is the lack of extensive experimental validation.
>
> We thank the reviewer for giving us the space to address this limitation.
>
> So far, we have performed the following additional experiments:
>
> 1) We showed that using the ESM-2 protein language model instead of ProtTrans as the backbone of Flexpert-3D, does not lead to improved performance. For details, please see our response to ‘W3’ raised by Reviewer fhjb.
> 2) To evaluate the quality of the generated protein sequences we ran an experiment with 700 randomly selected proteins from the CATH testset, for which we ran the Flexpert-Design predictions with the flexibility increasing instructions \=5 and the segment length S=50 (the same choice as in the experiment reported in Figure 4). For the Flexpert-Design predicted sequences, we ran the AlphaFold2 structure prediction (using the default settings of [localcolabfold](https://github.com/YoshitakaMo/localcolabfold)) and we compared the predicted structures of the predicted sequences with the ground truth structures at the level of $C_\alpha$ to compute the root mean square deviation between the structures (RMSD). Furthermore, we looked at the AF2 pLDDT confidence overall across the whole protein and separately inside and outside the engineered region (where we provided the flexibility increasing instructions). We present the results of the experiment in the table below (Table 9\) and we will include it in the Supplementary Material of the manuscript:
>
>
>
>    Table 9: Evaluation of the Flexpert-Design generated protein sequences using the AlphaFold2 pLDDT and     the alignment of the folded backbone structures to the ground truth backbone structures (at the $C_\alpha$ level).
>
>     | Model | RMSD  to Ground Truth ↓ (Å)  | ø pLDDT ↑ (whole protein) | ø pLDDT ↑ (engineered region) | ø pLDDT ↑ (outside engineered region) |
>     | :---- | :---- | :---- | :---- | :---- |
>     | ProteinMPNN | 2.00 | 0.77 | 0.81 | 0.73 |
>     | Flexpert-Design (no loss) | 3.31 | 0.61 | 0.60 | 0.62 |
>     | Flexpert-Design (ours) | 3.36 | 0.60 | 0.58 | 0.63 |
>
>
>
> We see that the Flexpert-Design generated sequences show lower recovery of the ground truth structure compared to ProteinMPNN, which could suggest that Flexpert-Design produces less stable structures or that it introduces notable conformational changes. A particularly significant drop in the pLDDT is in the engineered regions (which tended to be well-structured in the ground truth, as suggested by the high pLDDT for ProteinMPNN sequences), which might hint at the increased flexibility (which was shown by our predictor in Figure 4 and Table 5) causing the drop in pLDDT and increase in RMSD.
>
>
> We are currently working on more experiments in response to all of the 4 reviews we have received. We will post the new results later during the rebuttal period to address the reviewers' concerns.
>
> > **Questions:**
> > **Q1:** Given the relatively small ATLAS dataset, how does the predictor generalize to other protein datasets? Has the model been tested on proteins outside the ATLAS dataset to evaluate robustness?
>
> We thank the reviewer for the suggestion to test the generalizability to other datasets. For this purpose, we identified a recent dataset of MD simulations called mdCATH ([Mirachi et al. 2024](https://arxiv.org/abs/2407.14794)), which we are currently processing, and we will use it to test our models. The dataset was simulated at different temperatures (ranging from 320 K to 413 K) than the temperature used in the simulations of our trainset (300K, room temperature). In comparison to ATLAS, mdCATH explores non-physiological temperature ranges which provide information about protein unfolding.  We will provide the results of this experiment later during the rebuttal period.

---

> ### Author Response · Authors · 2024-11-22
> **Response to Reviewer i3SF (Part 2/2)**
>
> > **Q2:** I am wondering if the proteins generated by this model (due to the flexibility-aware nature of the model) show evidence of functional improvements (e.g., improved stability, interaction specificity), or is the flexibility change primarily theoretical?
>
> We thank the reviewer for this question. This depends on the particular protein—there are proteins where flexibility is believed to be directly linked to their function. It has been previously shown that the dynamic behavior of a bifunctional haloalkane dehalogenase/luciferase system affects the main activity output ([Schenkmayerova et al. 2021](https://www.nature.com/articles/s41929-022-00895-z), [Schenkmayerova et al. 2023](https://www.nature.com/articles/s41467-021-23450-z)). In protein engineering this has previously been addressed by a protocol to guide flexibility design by loop transplant ([Planas-Iglesias et al. 2022](https://academic.oup.com/nar/article/50/W1/W465/6570728)). Loop grafting in association to engineering protein dynamics has also been explored in beta-lactamases ([Gobeil et al. 2014](https://pubmed.ncbi.nlm.nih.gov/25200606/), [Park et al. 2006](https://www.science.org/doi/10.1126/science.1118953)), isomerases ([Romero-Rivera 2022](https://pubs.acs.org/doi/10.1021/jacsau.2c00063)), and tyrosyne phosphatases ([Crean et al. 2024](https://www.cambridge.org/core/journals/qrb-discovery/article/sequence-dynamics-function-relationships-in-protein-tyrosine-phosphatases/9A8C55E63F9852869D3A21B742705B79), [Shen et al. 2022](https://pubs.rsc.org/en/content/articlelanding/2022/sc/d2sc04135a)). In all these cases the enzymatic activity was modulated by flexibility in regions far from the catalytic site. Thus the engineering of their flexibility can lead to functional improvements. We have added more details on the motivation for engineering flexibility and how it can affect functional properties in the Introduction by extending the original statement:
>
> *“Proteins are highly dynamic biomolecules and the presence of flexible regions is critical for their biological function (Corbella et al., 2023; Lemay-St-Denis et al., 2022).”*
>
> with more concrete examples illustrating the importance of flexibility:
>
> *“In particular, fine-tuning conformational dynamics of loops close to the active site has been shown to be an important method for modulating substrate specificities ([Romero-Rivera et al. 2022](https://doi.org/10.1021/jacsau.2c00063)), turnover rates ([Crean et al. 2021](https://doi.org/10.1021/jacs.0c11806)), and pH dependency ([Shen et al. 2021](https://doi.org/10.1021/jacsau.1c00054)) of enzymes. In addition, the biological function of many proteins often requires that a small molecule is transported through their structures, for example via tunnels leading to the active site, whose dynamical properties are thus critical for protein function ([Jurcik et al. 2018](https://pubmed.ncbi.nlm.nih.gov/29741570/)). Modulating the flexibility of protein parts even far away from the active sites has been shown to be an efficient strategy for improving proteins ([Karamitros et al. 2022](https://doi.org/10.1073/pnas.2118979119)).”*
>
> The ultimate evidence of functional improvements will be a practical protein engineering use case with a wet lab validation, which we plan to do in the future. In particular, we plan to apply our method to increase flexibility in particular regions of a potential protein-based thrombolytic drug candidate and test the (theoretically expected) improvement of its thrombolytic activity in the wet lab.
>
> > **Q3:** The Flexpert loss function combines flexibility and sequence objectives with a mixing parameter, θ. How sensitive is the final model performance to different values of θ, and how was the optimal value chosen?
>
> We thank the reviewer for the question. Our aim was to preserve the original properties of ProteinMPNN (such as the sequence recovery) as much as possible while introducing to the model the awareness of flexibility. To balance out these different goals, we selected the value $\theta = 0.8$, as it was the maximal value (i.e., introducing the highest focus on flexibility) before the sequence recovery started dropping (see Table 7 in the Appendix). In general, the sequence recovery loss $L_{Seq}$ seemed more stable and had the tendency to dominate the training. The sensitivity to the value of $\theta$ was relatively low in the lower range of its values, so we had to push the balance more toward the flexibility loss (i.e., setting $\theta > 0.5$). A detailed discussion on balancing the value of parameter $\theta$ is provided in Appendix D.

---

> ### Author Response · Authors · 2024-11-25
> **Additional experiment: Evaluation of Flexpert-3D and Flexpert-Seq on mdCATH dataset**
>
> To complete our response to the reviewer,  we have performed an additional experiment evaluating our flexibility predictors over a new, recently introduced dataset (posted to ArXiv in July 2024). This addresses the reviewer's concern about the lack of experimental validation (‘W1’) and in particular the reviewer's question ‘Q1’:
>
> >Q1: Given the relatively small ATLAS dataset, how does the predictor generalize to other protein datasets? Has the model been tested on proteins outside the ATLAS dataset to evaluate robustness?
>
> To evaluate the generalization capability of Flexpert-3D and Flexpert-Seq predictors outside of the ATLAS dataset, we performed an additional experiment evaluating the predictors on the mdCATH dataset ([Mirarchi et al. 2024](https://arxiv.org/abs/2407.14794)).
>
> The recently introduced mdCATH dataset presents a challenging test of generalizability for our predictors trained on the ATLAS dataset. The mdCATH dataset is larger in terms of the number of data points (5398 compared to 1390 in ATLAS) and the simulated data correspond to individual protein domains (i.e., they are shorter proteins, on average with 137 residues compared to 236 residues in ATLAS). Furthermore, the simulation times in mdCATH are longer than in ATLAS (on average 464 ns for mdCATH compared to 100 ns for ATLAS) and the simulations were performed at different temperatures (320 K, 348 K, 379 K, 413 K, and 450 K) than the simulations in the ATLAS dataset (300 K). The results of the evaluation are presented in Table 10.
>
> Table 10: Evaluation of Flexpert-3D and Flexpert-Seq predictors on MD simulations from the mdCATH dataset. Both predictors were evaluated on the full dataset (5398 proteins, columns “Full data”) as well as on its subset excluding topologies present in the ATLAS training set (4013 proteins, columns “Topo. filtered”). The reported numbers are Pearson correlation coefficients between the model predictions and the RMSF from the dataset averaged over all the proteins in the dataset.
> |  | Flexpert-3D | Flexpert-Seq  |
> | :---- | :----  | :----  |
> | Simulation temperature | Full data / Topo. filtered | Full data / Topo. filtered |
> | $T = 320 K$ | 0.69 / 0.66 | 0.64 / 0.64 |
> | $T = 348 K$ | 0.64 / 0.63 | 0.63 / 0.63 |
> | $T = 379 K$ | 0.59 / 0.57 | 0.60 / 0.59 |
> | $T = 413 K$ | 0.49 / 0.48 | 0.52 / 0.52 |
> | $T = 450 K$ | 0.34 / 0.33 | 0.38 / 0.38 |
>
>
> Despite some performance drop with respect to the performance achieved on the ATLAS dataset (0.82 for Flexpert-3D and 0.76 for Flexpert-Seq), the performance of our predictors remains reasonable for the simulations performed at the temperature T = 320 K (0.66 for Flexpert-3D and 0.64 for Flexpert-Seq, for the topology filtered dataset). Considering that the temperature in this part of the mdCATH dataset is 20K away from the temperature of the simulations in the ATLAS training set, this appears as a reasonable generalization. We can notice that Flexpert-3D takes larger damage in its performance from the increased temperature of mdCATH simulations than Flexpert-Seq, suggesting that the sole reliance on sequence information makes the method more robust to the change in the simulation temperature. Note that the proteins are possibly getting unfolded in the high-temperature simulations of mdCATH, rendering non-physiological degrees of flexibility.
>
> We will add these additional experimental results to the Appendix and add a pointer to them in the main text.

---

### Official Review · Reviewer_fhjb · 2024-11-04

**Soundness:** 3
**Presentation:** 3
**Contribution:** 2
**Rating:** 6
**Confidence:** 4

**Summary:**

This paper proposes a method called Flexpert-Design to enable guidance of inverse folding models toward increased flexibility. The authors first quantify protein flexibility and identify data relevant to learning, then train flexibility predictors with protein language models. Finally, they fine-tune inverse folding models to steer it toward desired flexibility.

**Strengths:**

- **New field scenario**: Protein flexibility is rarely considered by current protein engineering methods.

- **Reasonable baseline settings**: The authors arange a reasonable set of flexibility-related baselines.

- **Quantified metric evaluation**: The authors apply several methods to quantify protein flexibility comprehensively.

**Weaknesses:**

- The technical novelty is somewhat limited. While the problem is important, the overall framework is simple and trivial.

- Flexpert-Design uses Gumbel-Softmax trick to enable back propagation. However, this estimation is not extremely accurate.

- Undiscussed choice consideration for the backbone.

- The experimental results are not comprehensive. Only flexibility-related metrics are shown. Besides, only ProteinMPNN is considered as the inverse folding model.

- Lack of evaluations on the overall protein quality (reliability, plausibility, etc.) for the sequences generated by Flexpert-Design. This can be an important proof of its practical application ability

- No clear explanation for the engagement of the proportion of flexibility-increasing mutations (What is its meaning? What properties and biological representations does it reflect?

**Questions:**

Here are some questions **from weaknesses**.

(1) Since estimation with Gumbel-Softmax trick is not extremely accurate, how does this method compared to other RL-based methods with the L\_flex as the reward?

(2) Why you choose ProtTrans as the backbone, do you try ESM series?

(3) Can the proposed method be applied to any inverse folding models?

And here are questions **about experiment details and considerations**.

(4) In Table 1, how do you match the RMSF of MD with other methods’ metrics (like plddt of AF2 and ESMFold2)? And in Sec. 3.1, you mention that MD is considered an accurate physics-based method. Is that a common sense? If so, why include the whole experiment in Table 1?

(5) How do you get the average PCC upper bound of 0.88? Is that for MD? Does it make sense to set an average value as the upper bound?

(6) How do you prove that the protein sequences designed by Flexpert-Design are natural (or useable) other than with good flexibility only? Since you mentioned that the loss function makes a balance between flexibility and sequence recovery, is it worth sacrificing the overall quality of proteins to enhance their flexibility?

---

> ### Author Response · Authors · 2024-11-22
> **Response to Reviewer fhjb (Part 1/4)**
>
> We thank the reviewer for their constructive feedback, which significantly helps us to improve our manuscript. Below, we address the individual points raised among ‘Weaknesses’ and ‘Questions’. We changed the bullet points and numbering from the original review to “W1”-”W6” and “Q1”-”Q6” to label weaknesses and questions, respectively. We then use these labels for cross-references in case of related questions and weaknesses.
>
> > **W1:** The technical novelty is somewhat limited. While the problem is important, the overall framework is simple and trivial.
>
> Thank you for letting us answer this concern and further clarify the contributions of this work, which may have not been explained clearly.
>
> First, we perform a comprehensive comparison of different methods for quantification of protein flexibility and identify relevant data for learning (Section 3). We believe this is an important contribution to enable progress in this area using learning-based methods. The significance of this contribution may reach beyond our specific method.
>
> Second, we leverage this data to design and learn a new protein flexibility predictor (Section 4). Here the technical challenges lie in (i) dealing with the low amount of training data and in (ii) finding how to effectively combine sequence and structure based information. Our innovations that address those challenges are: (i) leveraging and adapting, in a data efficient manner, a pre-trained protein language model to efficiently handle the sequence information and using the knowledge of Elastic Network Models to efficiently utilize the structural information and (ii) using a CNN-based trainable adaptor that allows combining structure and sequence information. Please see Figure 1 in the main paper.
>
> Third, we introduce a new method (Flexpert-Design) for fine-tuning a protein inverse folding model to make it steerable toward desired flexibility in specified amino acids (Section 5). Here the technical challenges lie in (i) finding a way how to provide the flexibility instruction as input to the already pre-trained inverse folding model and (ii) how to design a flexibility-aware loss function. Our technical innovations address those challenges by (i) augmenting the inverse folding model architecture to take the flexibility instructions as input, and by (ii) designing an architecture that uses our trained flexibility predictor (the second contribution, see the paragraph above) as part of the loss function.
>
> > **W2:** Flexpert-Design uses Gumbel-Softmax trick to enable back propagation. However, this estimation is not extremely accurate.
>
> We thank the reviewer for this point. Please see our response to ‘Q1’ below.
>
> > **W3:** Undiscussed choice consideration for the backbone.
>
> We thank the reviewer for bringing this up. We will detail the choice of the protein language model in the manuscript as suggested below. We chose ProtTrans based on the work studying the fine-tuning of protein language models by [Schmierler et al.](https://www.nature.com/articles/s41467-024-51844-2), which compares the fine-tuning of ProtTrans and ESM for various tasks and shows a very similar performance of ProtTrans and ESM-2 (the relevant details are in Section 1 of Supplementary Material of that paper).
>
> In the manuscript, we reflect this with the following modification (new text is in bold):
>
> "*To overcome the limitations coming from the small amount of annotated data (1390 proteins in the ATLAS dataset), we leverage the pre-trained protein language model ProtTrans (Elnaggar et al., 2022\) to obtain per-residue embeddings, on top of which we learn a linear layer to regress a single scalar value for each residue. **An alternative choice for the backbone protein language model could be the ESM-2 model, which was shown to perform similarly to ProtTrans in fine-tuning for diverse downstream tasks ([Schmierler et al. 2024](https://www.nature.com/articles/s41467-024-51844-2)).***"
>
> To answer the reviewer's question with a direct comparison of different backbones, we ran an initial experiment comparing under same conditions a Flexpert-3D model based on ProtTrans with a Flexpert-3D model based on ESM-2; the Pearson correlation coefficient between the predicted and the true RMSF was 0.79 in the case of ProtTrans and 0.77 in the case of ESM-2. If the reviewer is interested in this comparison, we can run a larger experiment on the computing infrastructure used in the manuscript in the coming days. However, based on the above referenced paper ([Schmierler et al. 2024](https://www.nature.com/articles/s41467-024-51844-2)), we believe that the choice between ProtTrans and ESM-2 is not critical.

---

> ### Author Response · Authors · 2024-11-22
> **Response to Reviewer fhjb (Part 2/4)**
>
> > **W4:** The experimental results are not comprehensive. Only flexibility-related metrics are shown. Besides, only ProteinMPNN is considered as the inverse folding model.
>
> We thank the reviewer for bringing up this point. We have run more experiments to make the experimental part more comprehensive: please see the experiment with the ESM backbone reported in the response to ‘W3’ above and the experiment on the protein quality reported in the response to ‘W5’ below, which also addresses the reviewer's point on having previously only reported the flexibility-related metrics. In addition, we will post more experimental results later during the rebuttal period to address suggestions made by other reviewers.
>
> Regarding the reviewer's point on our choice of ProteinMPNN, please see the response to ‘Q3’.
>
> > **W5:** Lack of evaluations on the overall protein quality (reliability, plausibility, etc.) for the sequences generated by Flexpert-Design. This can be an important proof of its practical application ability
>
> We thank the reviewer for the suggestion to evaluate the protein quality. To address this, we ran an experiment with 700 randomly selected proteins from the CATH testset, for which we ran the Flexpert-Design predictions with the flexibility increasing instructions $\tau=5$ and the segment length S=50 (the same choice as in the experiment reported in Figure 4). For the Flexpert-Design predicted sequences, we ran the AlphaFold2 structure prediction (using the default settings of [localcolabfold](https://github.com/YoshitakaMo/localcolabfold)) and we compared the predicted structures of the predicted sequences with the ground truth structures at the level of $C_\alpha$ to compute the root mean square deviation between the structures (RMSD). Furthermore, we looked at the AF2 pLDDT confidence overall across the whole protein and separately inside and outside the engineered region (where we provided the flexibility increasing instructions). We present the results of the experiment in the table below (Table 9\) and we will include it in the Appendix of the manuscript:
>
> Table 9: Evaluation of the Flexpert-Design generated protein sequences using the AlphaFold2 pLDDT and the alignment of the folded backbone structures to the ground truth backbone structures (at the $C_\alpha$ level).
>
> | Model | RMSD  to Ground Truth ↓ (Å)  | ø pLDDT ↑ (whole protein) | ø pLDDT ↑ (engineered region) | ø pLDDT ↑ (outside engineered region) |
> | :---- | :---- | :---- | :---- | :---- |
> | ProteinMPNN | 2.00 | 0.77 | 0.81 | 0.73 |
> | Flexpert-Design (no loss) | 3.31 | 0.61 | 0.60 | 0.62 |
> | Flexpert-Design (ours) | 3.36 | 0.60 | 0.58 | 0.63 |
>
> We see that the Flexpert-Design generated sequences show lower recovery of the ground truth structure compared to ProteinMPNN, which could suggest that Flexpert-Design produces less stable structures or that it introduces notable conformational changes. A particularly significant drop in the pLDDT is in the engineered regions (which tended to be well-structured in the ground truth, as suggested by the high pLDDT for ProteinMPNN sequences), which might hint at the increased flexibility (which was shown by our predictor in Figure 4 and Table 5) causing the drop in pLDDT and increase in RMSD.

---

> ### Author Response · Authors · 2024-11-22
> **Response to Reviewer fhjb (Part 3/4)**
>
> > **W6:** No clear explanation for the engagement of the proportion of flexibility-increasing mutations (What is its meaning? What properties and biological representations does it reflect?
>
> We thank the reviewer for this point. This metric was used to verify that the Flexpert-Design framework was successful in making the inverse folding model aware of the input flexibility instructions and that it enabled steering of the inverse folding model by the flexibility instructions. For example, in the case of Flexpert-Design, this metric tells us that if we instruct the model to increase the flexibility of a residue, it does so in 83% of cases. This metric might be useful for protein engineers who wish to increase flexibility in the protein of their interest, which is a very relevant task. To better provide the link between flexibility engineering and functional improvements of proteins, we enhance the motivation in the Introduction (which was also requested by Reviewer R4S7):
>
> *“In particular, fine-tuning conformational dynamics of loops close to the active site has been shown to be an important method for modulating substrate specificities ([Romero-Rivera et al. 2022](https://doi.org/10.1021/jacsau.2c00063)), turnover rates ([Crean et al. 2021](https://doi.org/10.1021/jacs.0c11806)), and pH dependency ([Shen et al. 2021](https://doi.org/10.1021/jacsau.1c00054)) of enzymes. In addition, the biological function of many proteins often requires that a small molecule is transported through their structures, for example via tunnels leading to the active site, whose dynamical properties are thus critical for protein function ([Jurcik et al. 2018](https://pubmed.ncbi.nlm.nih.gov/29741570/)). Modulating the flexibility of protein parts even far away from the active sites has been shown to be an efficient strategy for improving proteins ([Karamitros et al. 2022](https://doi.org/10.1073/pnas.2118979119)).”*
>
> > **Questions:**
> > **Here are some questions from weaknesses.**
>
> > **Q1:** Since estimation with Gumbel-Softmax trick is not extremely accurate, how does this method compared to other RL-based methods with the L\_flex as the reward?
>
> We thank the reviewer for this question. The Gumbel-Softmax trick indeed relies on approximation, which might limit the overall performance of our model. Our main point was to propose and demonstrate that it is possible to steer the inverse folding model by flexibility instructions to enable flexibility-aware protein sequence design. We were able to demonstrate the feasibility with our approach relying on the Gumbel-Softmax trick, but perhaps different techniques from Reinforcement Learning could lead to even better performance. To this end, we are considering the use of Direct Preference Optimization (DPO) ([Rafailov et al. 2023](https://arxiv.org/abs/2305.18290)) in our future work for the fine-tuning of the inverse folding model.
>
> > **Q2:** Why you choose ProtTrans as the backbone, do you try ESM series?
>
> We thank the reviewer for the question, please see our response to ‘W3’ above.
>
> > **Q3:** Can the proposed method be applied to any inverse folding models?
>
> We thank the reviewer for this question. The Flexpert-Design framework modifies the training of the inverse folding model in two ways. Firstly, it modifies the input of the inverse folding model to be able to receive the flexibility instructions. In our work, this is specific to models with Graph Neural Network (GNN) architecture of the encoder, such as ProteinMPNN, as the flexibility instructions are used to initialize the node features of the graph-encoded input backbone structure. Secondly, the Flexpert-Design framework introduces the training with the L\_Flex loss, which could be applied to any inverse folding model.
>
> Therefore, our method can be applied as is to any inverse folding model with a GNN encoder, but it also could be applied to any inverse folding model if the model was adapted to receive flexibility instructions on the input.
>
> We decided to focus on the ProteinMPNN model because it is the best-established model in practice. This choice was acknowledged by Reviewer DNnV.

---

> ### Author Response · Authors · 2024-11-22
> **Response to Reviewer fhjb (Part 4/4)**
>
> > **Q4:** In Table 1, how do you match the RMSF of MD with other methods’ metrics (like plddt of AF2 and ESMFold2)? And in Sec. 3.1, you mention that MD is considered an accurate physics-based method. Is that a common sense? If so, why include the whole experiment in Table 1?
>
> We thank the reviewer for the questions. We match RMSF from MD simulations with the confidence of the structure prediction models by taking the 1-pLDDT value. For details on the quantification of flexibility by pLDDT, please see Sec. 3.1 and our response to question ‘Q2’ from Reviewer R4S7. Reviewer R4S7 pointed out that it is counterintuitive for “pLDDT uncertainty” to correlate negatively with flexibility. To this end we provide the following clarification:
>
> We have modified the manuscript to refer to the pLDDT score as “pLDDT confidence” (rather than "pLDDT uncertainty"). The pLDDT score, as defined for example in AF2, is in the interval \[0,1\], where the 0 and 1 values correspond to, respectively, absolutely uncertain and fully confident predictions of the structure. While a low confidence might be due to the error of the model (for example because the model has not seen relevant examples in its training set), it can also be caused by uncertainty in the ground truth structure if that structure is flexible. A higher pLDDT score (i.e., a higher pLDDT confidence) can thus indicate a more rigid structure and lower flexibility.
>
> Regarding the point on choosing MD simulations as the ground truth and the purpose of Table 1, it was not clear whether MD simulations (as an accurate physics-based method) or B-factors (as a product of an experimental method) should be used as the ground truth. Table 1 helped us understand that MD simulations are easier to approximate by other methods than B-factors. We further verified that RMSF from MD simulations are easier to learn from when training Flexpert-Seq on RMSF from MD simulations (Table 3\) than on B-factors (Table 6 in Appendix B). Note that MD simulations are very computationally demanding, motivating the use of faster alternatives, which we also tried to survey by Table 1 and finally tackle by the development of Flexpert-Seq and Flexpert-3D.
>
> > **Q5:** How do you get the average PCC upper bound of 0.88? Is that for MD? Does it make sense to set an average value as the upper bound?
>
> We thank the reviewer for the question. While we consider the MD simulation to be the most accurate method available to estimate the flexibility, it still contains some noise, which we tried to quantify. Therefore, we used the fact that all the MD simulations in the ATLAS dataset were run three times, with different starting velocities assigned from a Boltzman distribution. We computed the correlation of RMSFs between the different replicas, which gave us the PCC \= 0.88 on average across the ATLAS dataset. We use the MD-generated ground truth in the training, but since even newly initialized MD itself can only recover the number with PCC \= 0.88, we do not expect any other (non-overfitted) method to perform better than with PCC \= 0.88, so we consider it the (indicative) upper bound.
>
> We believe that the use of an average upper bound makes sense when applied to the average value computed for the same dataset. The reviewer is right that this upper bound cannot be applied to individual predictions. We have clarified this in the manuscript by changing the original sentence:
>
> *“We consider this number to be the upper bound for the correlation of any flexibility predictor trained on the ATLAS dataset.”*
>
> to
>
> *“We consider this number to be the **indicative** upper bound for the **average** correlation of any flexibility predictor trained **and evaluated** on the ATLAS dataset.”*
>
> > **Q6:** How do you prove that the protein sequences designed by Flexpert-Design are natural (or useable) other than with good flexibility only? Since you mentioned that the loss function makes a balance between flexibility and sequence recovery, is it worth sacrificing the overall quality of proteins to enhance their flexibility?
>
> We thank the reviewer very much for this question, please see our response to ‘W5’ above.

---

> > ### Comment · Reviewer_fhjb · 2024-11-28
> >
> > Many thanks for the detailed response to my questions, which has addressed my major concern. I have increased my score accordingly.

---

> > > ### Author Response · Authors · 2024-11-29
> > >
> > > We thank the reviewer very much for acknowledging our response and for their comprehensive review, which helped us improve our manuscript.

---

### Author Response · Authors · 2024-11-22

Dear reviewers,

Thank you very much for your suggestions and insightful comments. We have addressed them in the individual responses to your reviews.

To answer your questions, we have run experiments to answer the questions about the quality of the proteins generated by Flexpert-Design and to compare the chosen protein language model ProtTrans with ESM-2 as an alternative.

By the end of the rebuttal period we also plan to upload the modified pdf file, reflecting many of your suggestions on writing and presentation. These suggestions have been so far addressed in the responses to your reviews.

---

### Author Response · Authors · 2024-11-29

Dear reviewers,

Thank you very much for your insightful comments and constructive suggestions, which helped us to considerably improve our manuscript.

Based on your suggestions, we have improved the presentation of the manuscript by adding several clarifying comments and strengthening the motivation for the studied problem.

Furthermore, during the discussion period, we have performed 5 additional experiments addressing your points:

1. Evaluation of the structure preservation in Flexpert-Design generated sequences using AlphaFold 2 (see Appendix I).
2. Evaluation of Flexpert-Seq and Flexpert-3D on a new independent dataset mdCATH (Appendix F).
3. Qualitative evaluation of Flexpert-Seq and Flexpert-3D (Appendix G).
4. Retraining of Flexpert-3D using ESM-2 as the backbone language model (Response to Reviewer fhjb (Part 1/4))
5. Evaluation of a domain knowledge-based baseline for flexibility engineering (Response to Reviewer DNnV - “Qualitative examination of Flexpert-Seq and Flexpert-3D and a new baseline for Flexpert-Design.”)

Previously, we addressed your reviews in our responses in the discussion. We have now updated the PDF of our manuscript to reflect the changes, with all new text highlighted in blue. The main changes are as follows:

1. We have strengthened the motivation for engineering protein flexibility by providing references to concrete use cases where this strategy has led to experimentally validated improvements in the function of several proteins.
2. We have merged the former Sections 4 and 5 into a single section describing the methods (now Section 4).
3. Due to the page limit, we have moved the former Table 2 (now Table 6) and the accompanying discussion to Appendix E. We have also moved the former Figure 3 (now Figure 8) and the accompanying discussion to Appendix J.
4. In addition to the Appendices E and J, we have added 4 brand new sections to the Appendix (Appendices B, F, G, and I). While Appendix B provides more details on the calculation of root mean square fluctuations from MD simulations and Elastic Network Models, Appendices F, G, and I report on 3 of the 5 new experiments performed during the discussion period.



We have tried to address all points raised in the reviews. If any of the reviewers have further questions, we will be happy to answer them during the extended discussion period.

Thank you very much again for your valuable feedback.

---

### Meta-Review · Area_Chair_iieQ · 2024-12-21

**Metareview:**

This paper works towards modifying protein design models (e.g., inverse folding models) to be capable of producing proteins with higher flexibility. The authors propose metrics for measuring flexibility, and then fine tune a protein language model (ProtTrans) with LoRA to predict flexibility from the relatively small amount of annotated data available. They demonstrate the success of their method by showing that, under these metrics, they are able to produce more flexible proteins than standalone ProteinMPNN. The reviewers mostly agree in the end that the problem is interesting, not often considered and well motivated, and the authors do a thorough job of setting up evaluation metrics, creating a supervised fine tuned model for the problem, and then augmenting generative protein design models.

**Additional Comments On Reviewer Discussion:**

A common question raised was the relatively small structures being evaluated, to which the authors have supplemented the ATLAS results with additional results on the mdCATH, with more results to be completed by camera ready. The authors have also performed better ablation of whether their protein designs alter the structure using AlphaFold 2 (new Appendix I). The author feedback during the review period explicitly satisfied three of the reviewers who have raised their scores accordingly.

---

### Decision · Program_Chairs · 2025-01-22

Accept (Poster)